



# Radiometric calibration of a non-imaging airborne spectrometer to measure the Greenland Ice Sheet surface

Christopher J. Crawford[1,2,3], Jeannette van den Bosch[4], Kelly M. Brunt[1,2], Milton G. Hom[5,6,7] John W.
Cooper[5,6,8], David J. Harding[6], James J. Butler[6,8], Philip W. Dabney[9], Thomas A. Neumann[2], Craig S.
Cleckner[10] , Thorsten Markus[2]

[1]Earth System Science Interdisciplinary Center, University of Maryland, 5825 University Research Court #4001, College Park,
Maryland 20704, USA
[2]Cryospheric Sciences Laboratory (Code 615), NASA Goddard Space Flight Center, 8800 Greenbelt Road, Greenbelt,
Maryland 20771, USA
[3]Arctic Slope Regional Corporation Federal InuTeq, contractor to the U.S. Geological Survey Earth Resources Observation
and Science Center, Science and Applications Branch, 47914 252nd Street, Sioux Falls, South Dakota, 57198, USA
[4]Air Force Research Laboratory, Battlespace Surveillance Innovation Branch, Kirtland Air Force Base, New Mexico 87117,
USA
[5]Science Systems and Applications Inc., 10210 Greenbelt Road #600, Landham, Maryland 20706, USA
[6]Biospheric Sciences Laboratory (Code 618), NASA Goddard Space Flight Center, 8800 Greenbelt Road, Greenbelt, Maryland
20771, USA
[7]Biospheric Optics Laboratory (Code 618), NASA Goddard Space Flight Center, 8800 Greenbelt Road, Greenbelt, Maryland
20771, USA
[8]Radiometric Calibration Laboratory (Code 618), NASA Goddard Space Flight Center, 8800 Greenbelt Road, Greenbelt,
Maryland 20771, USA
[9]Laser Remote Sensing Laboratory (Code 694), NASA Goddard Space Flight Center, 8800 Greenbelt Road, Greenbelt,
Maryland 20771, USA
[10]Research Services Division (Code D1), NASA Langley Research Center, 1 NASA Drive, Hampton, Virginia 23666, USA

*Correspondence to*: Christopher J. Crawford (cjcrawford@contractor.usgs.gov)

**Abstract.** Methods to radiometrically calibrate a non-imaging airborne visible-to-shortwave infrared (VSWIR) spectrometer
to measure the Greenland Ice Sheet surface are presented. Airborne VSWIR measurement performance is then benchmarked
for bright Greenland ice and dark bare rock/soil targets using the MODerate resolution atmospheric TRANsmission
(MODTRAN) radiative transfer code (version 6.0), and a coincident Landsat 8 Operational Land Imager (OLI) acquisition on
29 July 2015 during an in-flight radiometric calibration experiment. Airborne remote sensing flights were carried out in
northwestern Greenland in preparation for the Ice, Cloud and land Elevation Satellite 2 (ICESat-2) laser altimeter mission.
Nine science flights were conducted over the Greenland Ice Sheet, sea ice, and open ocean water. The campaign's primary
purpose was to correlate green laser pulse penetration into snow and ice with spectroscopic derived surface properties. An
experimental airborne instrument configuration that included a nadir viewing (downward looking at the surface) non-imaging
Analytical Spectral Devices Inc. (ASD) spectrometer that measured at-sensor upwelling VSWIR (0.35 to 2.5 µm) spectral
radiance (Watts/m$^{-2}$/sr$^{-1}$/nm$^{-1}$) in the two color Slope Imaging Multi-polarization Photon-Counting Lidar's (SIMPL) ground
Instantaneous Field-of-View, and a zenith viewing (upward looking at the sky) ASD spectrometer that measured at-sensor
VSWIR spectral irradiance (Watts/m$^{-2}$/nm$^{-1}$) was flown. Rigorous radiometric calibration procedures for laboratory, in-flight,



and field environments are described in detail to achieve a targeted at-sensor VSWIR measurement requirement of within 5% to support calibration/validation (cal/val) efforts and geophysical science algorithm development. Our MODTRAN simulations for the 29 July flight line over dark and bright targets indicate that the nadir viewing airborne VSWIR spectrometer achieved an at-sensor spectral radiance measurement accuracy of between 0.6 and 4.7% for VSWIR wavelengths (0.4 to 2.0

µm) with atmospheric transmittance greater than 80%. At-sensor MODTRAN simulations for Landsat 8 OLI relative spectral response functions suggest that OLI is measuring 6 to 16% more at-sensor top-of-atmosphere (TOA) spectral radiance from the Greenland Ice Sheet surface than was observed from the nadir viewing airborne VSWIR spectrometer. While more investigation is required to convert airborne at-sensor VSWIR spectral radiance into atmospherically-corrected airborne surface reflectance, it is expected that airborne science flight data products will contribute to spectroscopic determination of

Greenland Ice Sheet surface properties to improve understanding of their potential influence on ICESat-2 measurements.

## 1. Introduction

Calibrated spectral radiance measurements from multispectral and imaging spectrometer instruments are a baseline requirement for producing geophysical data products that can be used to study Earth's land, ice, water, and atmospheric environments (Green, 1998;Green et al., 2006;King et al., 1996;Schaepman-Strub et al., 2006;Thome, 2001;Vane et al., 1993).

Optical instrument calibration is based on a traceable radiance standard determined by the National Institute of Standards and Technology (NIST) in the United States for example, where radiance measurements are collected from a stable illumination source in a controlled laboratory environment (Chrien et al., 1990;Schaepman and Dangel, 2000;Strobl et al., 1997;Tansock et al., 2015;Parr and Datla, 2001). Using this stable NIST traceable source, periodic assessments of an optical instrument's response are made to monitor its long-term repeatability, mechanical functionality, and responsivity to variable light intensities.

While radiometric calibration is fundamental to spectral instrument data acquisition, this is especially critical for missions bound for deployments in Polar Regions because the range of measured snow, ice and liquid water surfaces spans the entire solar spectrum dynamic range. For airborne missions, precise and accurate pre-flight, in-flight, and post-flight calibration procedures are therefore of paramount importance to achieve targeted instrument stability and measurement accuracy requirements. Commitment to characterize instrumentation, instrument foreoptics, and supporting aircraft hardware during

pre- and post-airborne mission timelines helps to produce geophysical measurements in which uncertainty has been constrained and fully calibrated data products are available to support algorithm development and science applications.

In this paper, we describe laboratory, in-flight, and field radiometric calibration procedures necessary to obtain science quality measurements from a visible-to-shortwave infrared (VSWIR) non-imaging airborne spectrometer. We used the MODerate

resolution atmospheric TRANsmission (MODTRAN) code version 6.0 (Berk et al., 2005) to benchmark the at-sensor measurement performance of the nadir viewing airborne VSWIR spectrometer over bright Greenland ice and dark bare rock/soil targets during a 29 July 2015 in-flight radiometric calibration experiment. Two non-imaging airborne VSWIR



spectrometers were flown as a part of the Slope Imaging Multi-polarization Photon-Counting Lidar (SIMPL)/Advanced Visible Infrared Imaging Spectrometer-Next Generation (AVIRIS-NG) 2015 airborne campaign to northwest Greenland in July and August 2015 (Brunt et al., 2015). The nadir viewing VSWIR spectrometer's objective was to acquire non-imaging profile measurements of snow, ice and liquid water light scattering and absorption, and the zenith viewing VSWIR spectrometer's

objective was to characterize sky conditions during nine science flights. The campaign was conducted in support of the Ice, Cloud and land Elevation Satellite 2 (ICESat-2) mission scheduled for launch in 2018. ICESat-2, a follow-on laser altimeter mission to ICESat (Schutz et al., 2005;Zwally, 2002), will continue measurements of ice sheet elevation and change, sea ice thickness, ocean surface height, land topography, vegetation height and structure and atmospheric clouds and aerosols. The Geoscience Laser Altimeter System (GLAS) (Abshire et al., 2005) on the ICESat mission used a traditional single-beam, near-

infrared (1064 nm, NIR), analog waveform method for the surface altimetry measurements. The Advanced Topographic Laser Altimeter System (ATLAS) (Abdalati and Zwally, 2010;Markus et al., 2017) on ICESat-2 will use a more efficient measurement producing multiple beams using a green (532 nm) micropulse, photon counting approach.

In order to prepare for the ICESat-2 mission, the Greenland campaign was conducted to better understand how ATLAS will

represent the height, roughness and topography of snow and ice surfaces to determine the spatial extent, and potentially the depth, of melt water on the ice sheet and sea ice surface. Four instruments were flown, two of which included non-imaging airborne VSWIR spectrometers. The dual airborne VSWIR spectrometer integration was considered experimental to the Greenland campaign's overall mission objective. The non-imaging airborne VSWIR spectrometers and the Slope Imaging Multi-polarization Photon-Counting lidar (SIMPL) (Dabney et al., 2010;Harding et al., 2011) were flown together on the

NASA Langley Research Center (LaRC) King Air (UC-12B). SIMPL uses a micropulse, photon counting, multi-beam measurement like that of ATLAS, but provides added information about light scattering by using co-aligned green and NIR laser pulses and a measure of pulse depolarization. AVIRIS-NG (Hamlin et al., 2010) was flown on a King Air (C-12) operated by Dynamic Aviation. Snow radiative transfer modeling (Aoki et al., 2000;Bohren and Barkstrom, 1974;Libois et al., 2014;Libois et al., 2013;Painter and Dozier, 2004a;Picard et al., 2009;Warren, 1982;Wiscombe and Warren,

1980;Kokhanovsky and Zege, 2004) and VSWIR spectroscopy studies has shown that optical snow surface reflectivity is most sensitive to concentrations of light absorbing impurities (e.g., dust, soot and black carbon containments) at visible wavelengths (Aoki et al., 2000;Dozier et al., 2009;Painter et al., 2007;Painter et al., 2009;Painter et al., 2013;Warren, 2013;Warren and Wiscombe, 1980), whereas effective snow surface grain size is a measure of melt state which can be quantified by exploiting the position, depth and shape of spectral absorption features within infrared wavelengths (Clark and Roush, 1984;Dang et al.,

2016;Dozier and Painter, 2004;Gardner and Sharp, 2010;Green et al., 2006;Libois et al., 2014;Libois et al., 2013;Nolin and Dozier, 2000;Painter et al., 2009;Painter et al., 1998;Warren et al., 2006;Wiscombe and Warren, 1980).

Because the ATLAS green laser pulses may penetrate into snow and ice, to a significant depth to cause surface height measurements to be biased low, the primary objective of the SIMPL/AVIRIS-NG 2015 Greenland campaign was to obtain the





necessary geophysical measurements to enable the ICESat-2 project to determine if green light depth of penetration, measured by SIMPL, is correlated with surface grain size, contaminant and/or wetness properties determined using VSWIR spectra. A comparison of green laser pulse shape broadening caused by volume scattering in snow, ice and liquid water, as compared to NIR pulses that only undergo surface scattering, provides the measurement of penetration depth. If that depth is correlated

with any particular surface property, changes in those properties seasonally and/or inter-annually could potentially cause bias in rates of ice sheet elevation change from ICESat-2 retrievals. The nadir viewing VSWIR spectrometer optical head was mounted inside SIMPL and their Instantaneous-Field-of-Views (IFOVs) were aligned to ensure the spectroscopic and altimetry profile measurements were co-incident, observing the same surface location at the same time through the same atmospheric column. AVIRIS-NG followed the SIMPL flight path at a higher altitude and trailing by about 15 minutes. Flying with

AVIRIS-NG was important because its estimations of grain size, contaminant concentrations, and wetness are relatively mature and by imaging a swath, it provides information about the spatial variability of these surface properties.

The non-imaging airborne VSWIR spectrometer integration on the NASA LaRC King Air included a nadir veiwing VSWIR spectrometer measuring at-sensor upwelling spectral radiance (Watts/m$^{-2}$/sr$^{-1}$/nm$^{-1}$, where sr is the FOV full angle), and an

zenith viewing VSWIR spectrometer measuring at-sensor downwelling spectral irradiance (Watts/m$^{-2}$/nm$^{-1}$). We prescribed MODTRAN with a sub-arctic standard aerosol profile, maritime radiation transport model, top-of-atmosphere (TOA) solar irradiance, CIMEL atmospheric measurements as part of AErosol RObotic Network (AERONET) (Holben et al., 1998), spectral response functions, and solar and line-of-sight (LOS) geometries. We simulated at-sensor spectral radiance for the nadir viewing airborne VSWIR spectrometer over bright Greenland ice and dark bare rock/soil targets to determine whether

airborne measurement performance was within the targeted 5% requirement. We selected flight segments from the 29 July in-flight radiometric calibration experiment that was intended to optimize the nadir viewing VSWIR spectrometer's visible-near infrared (VNIR) integration time and shortwave infrared (SWIR) gains across the full solar spectrum dynamic range. Along the northern portion of the NASA LaRC 29 July flight line over the Greenland Ice Sheet interior, Landsat 8 Operational Land Imager (OLI) acquired a coincident multispectral image.

We exploited this Landsat 8 OLI image acquisition by simulating at-sensor TOA spectral radiance for OLI using the identical MODTRAN benchmarking approach as constructed for the nadir viewing airborne VSWIR spectrometer. Because Landsat is the gold standard for optical satellite cal/val (Markham and Helder, 2012), we felt it was important to evaluate the nadir viewing airborne VSWIR spectrometer's bright Greenland ice measurement performance along with Landsat 8 OLI as an additional

comparison step. Landsat's capabilities to measure Polar Regions since the launch of Landsat 8 in February 2013 has been unprecedented because of onboard instrument performance and changes to its long-term acquisition plan that includes imaging of all sunlight land and coastal regions greater than 5° solar elevation. Imaging higher latitudes and polar ice sheets in solar-reflected wavelengths presents several challenges that are a result of low solar illumination angles, surface bidirectional reflectance distribution function (BRDF) effects (Aoki et al., 2000;Hudson et al., 2006), longer path length and greater



atmospheric refraction, and persistent cloudiness with cloud shadows cast on the ice sheet (Choi and Bindschadler, 2004;Hudson and Warren, 2007). Yet, because Landsat's orbital tracks converge at the poles, swath imaging side lap results in much higher temporal imaging frequency than tropical and middle latitude regions.

The specific objectives of this paper are to: (1) describe the non-imaging airborne VSWIR spectrometer integration and radiometric calibration procedures for pre-flight, in-flight, and post-flight timeframes; (2) describe the equations necessary to calculate the nadir viewing VSWIR spectrometer ground IFOV footprint; (3) characterize downwelling VSWIR spectral irradiance measurements to screen out unusable data and support atmospheric compensation modeling; and (4) benchmark the nadir viewing airborne VSWIR spectrometer's at-sensor measurement performance over bright Greenland ice and dark bare
rock/soil targets using MODTRAN and a coincident Landsat 8 OLI image acquisition.

## 2. Non-Imaging Airborne VSWIR Spectrometry

### 2.1 VSWIR Spectrometer Description

The non-imaging VSWIR spectrometers belong to the Earth Sciences Division (Code 610) at NASA's Goddard Space Flight Center (GSFC). The nadir viewing VSWIR spectrometer is a full range ASD FieldSpec Pro model maintained by the Code
618 Optics Laboratory. The zenith viewing VSWIR spectrometer is a full range ASD FieldSpec 3 model maintained by the Code 618 Radiometric Calibration Laboratory (RCL). Both instruments have visible-to-near infrared (VNIR) detectors (i.e., 350-1000 nm wavelength) with a Si photodiode array overlaid with an order-sorting filter. The shortwave infrared (SWIR) detectors (i.e., 1001-1800 and 1801-2500 nm wavelengths) are thermoelectrically cooled InGaAs photodiodes.

### 2.2 VSWIR Spectrometer Integration with SIMPL

The VSWIR spectrometers were mounted and secured on aluminium racks within the UC-12B fuselage at LaRC. The nadir viewing VSWIR spectrometer 1° foreoptic was mounted and secured within the SIMPL housing centered over a flat BK7 optical window. The fiber optic cable was connected to the nadir viewing VSWIR spectrometer, and a parallel port cable was used to communicate with the instrument control laptop. The zenith viewing VSWIR spectrometer remote cosine receptor (RCR) was mounted on top of the aircraft in an external enclosure with a flat BK7 optical window. The enclosure, referred to
hereinafter as the 'OrangeCan', was mounted in a zenith position and bolted and sealed to the aircraft roof to maintain cabin pressure during flight (Figure 1). The fiber optic cable was connected to the zenith viewing instrument through a small communication port, and an Ethernet cable was used to communicate with the instrument control laptop.

The IFOV alignment between SIMPL and the nadir viewing VSWIR spectrometer 1° foreoptic was confirmed using a ground
test procedure in an aircraft hangar with low light conditions. The SIMPL downward-directed laser beams were turned to a horizontal path and directed at a white reference target. The SIMPL laser transmitter produces four laser beams that are



distributed perpendicular to the aircraft flight direction. The locations of the four visible green laser spots on the target were identified. The center of the nadir viewing VSWIR spectrometer FOV was determined by translating a white light source across the target, with its pointing direction parallel to the laser beams. The FOV center position was established by real-time observation of the VSWIR spectrometer's peak response to the light source. At the nominal flight altitude of 2,500 m above

ground level (AGL), the 1° foreoptic IFOV produces a 44 m diameter ground sampling footprint. The SIMPL 0.4° spread of the beams and 0.007° beam divergence produces 0.3 m diameter ground spots distributed 20 m cross-track. We determined that the beams are located at the trailing edge of the nadir viewing VSWIR spectrometer's IFOV with the footprints displaced approximately 10 m to the right of the IFOV center.

### 2.3 VSWIR Spectrometer Measurements

Instrument control laptops for both VSWIR spectrometers required manual operation to initialize the appropriate instrument control software. The spectroscopic measurement interval for both nadir and zenith veiwing VSWIR spectrometers was set to one second (i.e., fastest programmable measurement time), and the integration time for the VNIR detector and gain setting for SWIR1 and SWIR2 detectors remained fixed for the entire airborne mission that included a dark current subtraction during each flight. The scan time for SWIR1 and SWIR2 detectors is ~220 milliseconds, thus, the total time between measurements

included the VNIR integration time, SWIR1 and SWIR1 scan time, and file save time. The VSWIR measurements were time-tagged recorded at a temporal integration interval of ~1 second, and an along-track length scale of ~100 meters.

Nadir and zenith viewing spectroscopic measurements during each flight were stored as 16-bit raw digital counts for the 0.35 to 2.5 µm VSWIR spectral range. Raw counts from both instruments were converted to at-sensor upwelling VSWIR spectral

radiance and at-sensor downwelling VSWIR spectral irradiance using calibration coefficients. Parabolic corrections were applied to each measurement to splice together VNIR, SWIR1, and SWIR2 detectors. Each upwelling spectral radiance and downwelling spectral irradiance measurement had a Universal Time Coordinated (UTC) timestamp that was synchronized with Applanix GPS time and geolocation during flight.

### 3. VSWIR Spectrometer Radiometric Calibration

### 3.1 Pre-Flight Laboratory Calibration Procedures

### 3.1.1 Nadir Viewing VSWIR Spectrometer

FieldSpec Pro linearity and repeatability tests were conducted using a NIST traceable illumination source and integrating sphere in the NASA/GSFC Code 618 Optics Laboratory. To check the instrument's linearity, the initial NIST source calibration strategy for VNIR and SWIR detectors was optimized, and then the VNIR integration time and SWIR1/2 gains were increased

by 50% to test the instrument's response (Figure 2). Bare fiber (25° IFOV) radiance measurements were captured from the integrating sphere output where the fiber optic tip was centered in front of the aperture. To assess instrument repeatability over



time, bare fiber radiance is periodically captured using identical linearity test procedures. FieldSpec Pro stability was determined to be less than 2% for VNIR, SWIR1, and SWIR2 detectors for pre- and post-flight timeframes (Figure 2). Spectral calibration of the FieldSpec Pro VNIR and SWIR detectors is routinely conducted using Mercury and Argon signatures with a resulting wavelength precision of less than 2% for 1 nm resolution.

## 3.1.2 Zenith Viewing VSWIR Spectrometer

The FieldSpec 3 linearity test was conducted using the same procedures as the FieldSpec Pro instrument (Figure 3). Prior to aircraft integration, ASD Inc. (a PANalytical company) conducted routine instrument maintenance and spectral calibration checks on the FieldSpec 3. The FieldSpec 3 was determined to be stable with a wavelength precision of less than 2% for 1 nm resolution. Although longer term information on FieldSpec 3 repeatability was unavailable, a cross-calibration between FieldSpec Pro and FieldSpec 3 bare fiber radiance using the NASA/GSFC Code 618 Optics Laboratory NIST traceable source indicated that the between VSWIR spectrometer response difference was within 2% for wavelengths between 0.5 to 2.0 µm (Figure 3).

## 3.1.3 Optical Window Transmission and Measurement Requirements

Optical window light transmittance is wavelength dependent. The BK7 optical window, procured from ESCO Optics, was mounted in the OrangeCan right above the RCR optic. We measured BK7 window transmittance using the FieldSpec Pro and NIST traceable source. The optical window was mounted and centered in front of the integrating sphere aperture. The VSWIR spectrometer fiber optic tip was mounted and placed in front of the optical window. We captured radiance measurements at top, right, bottom, left, and center window positions to fully assess transmission. We averaged optical window measurements and compared with window-free integrating sphere radiance to derive wavelength-dependent radiance loss due to window transmissivity (Figure 4). The nadir viewing VSWIR spectrometer BK7 optical window for the LaRC UC-12B aircraft was procured from Comso Optics Inc. Transmittance for this optical window was determined to be greater than 90% for wavelengths between 0.34 and 2.2 µm per manufacture specifications.

Optical window transmission specifications provided a realistic baseline for upwelling (downwelling) VSWIR spectral radiance (irradiance) measurement requirements because both nadir and zenith viewing VSWIR spectrometer stability was determined to be less than 2% using the NIST traceable source. Upwelling, at-sensor VSWIR spectral radiance measurement accuracy for wavelengths between 0.4 - 2.0 µm was determined to be within 5% for the nadir viewing VSWIR spectrometer. Downwelling, at-sensor VSWIR spectral irradiance measurement accuracy for wavelengths between 0.4 - 2.0 µm was determined to be within 4% for the zenith viewing VSWIR spectrometer. For both nadir and zenith viewing VSWIR spectrometers, measurement accuracy for wavelengths between 2.0 - 2.5 µm was between 5 and 13%, and primarily attributable to radiance loss due to optical window transmissivity.



### 3.2 In-Flight Calibration Procedures

### 3.2.1 Nadir Viewing VSWIR Spectrometer

The 29 July flight over the Greenland Ice Sheet interior was used for in-flight nadir viewing VSWIR spectrometer radiometric calibration to measure at-sensor spectral radiance during the entire airborne mission. The range of measured snow, ice and liquid water surfaces during this flight covered a full-reflected solar spectrum dynamic range from bright land ice with coarse snow grains, to darker bare rock/soil, to dark open ocean water. The in-flight radiance calibration strategy was iteratively optimized for the VNIR detector integration time and SWIR1/2 detector gain setting to avoid at-sensor spectral radiance saturation when flying across strong snow, ice, and liquid water surface gradients. We chose to optimize the at-sensor spectral radiance calibration strategy over interior Greenland ice with a probable dry snow layer, while under near clear-sky solar illumination conditions. This in-flight radiometric calibration strategy was designed to constrain the upper limits of at-sensor spectral radiance over a bright target, while recovering as much low radiance signal as possible over dark targets within the LOS during summertime solar zenith angles (SZAs).

Even though the nadir viewing VSWIR spectrometer was mounted with a nadir IFOV and the NASA LaRC UC-12B was in a stable horizontal position during flight, we note two specific in-flight caveats that are inherent to airborne measurements. First, in-flight inclination can subtly impact the nadir viewing geometry in that it can be difficult to determine exactly how short-term atmospheric turbulence and/or aircraft positional change influences the BRDF of the measured surface anisotropy within the IFOV. The SIMPL instrument aboard the NASA LaRC UC-12B recorded inclination during flight and could be used to constrain this measurement artefact in a post-processing mode. We determined this to not be significant relative to the at-sensor spectral radiance measurement accuracy requirement discussed in Section 3.1.3.

Second, snow and ice surfaces have an anisotropic signature dominated by forward scattering (Aoki et al., 2000;Leshkevich and Deering, 1990;Painter and Dozier, 2004b;Schaepman-Strub et al., 2006), and can also be highly specular during melt (Leshkevich and Deering, 1990;Mullen and Warren, 1988). Thus, the aircraft along-track LOS within the flight path is important to reconcile relative to direct path solar illumination geometry. If the aircraft LOS is generally perpendicular to the direct path solar principal plane, then airborne measured snow and ice at-sensor radiance will not suffer greatly from an angular scattering bias. However, if the aircraft LOS is parallel or near-parallel to the direct path solar principal plane, then either a BRDF correction must be applied or caution must be exerted prior to interpretation of airborne reflectance. Flying underneath homogenous cloud layers results in an isotropic assumption where surface scattering is not dependent on direction (Hudson and Warren, 2007). Science flights that have the potential for an along-track LOS scattering bias will be flagged in the measurement metadata.



### 3.2.2 Zenith Viewing VSWIR Spectrometer

In-flight radiometric calibration of the zenith viewing VSWIR spectrometer to measure at-sensor spectral irradiance was also conducted during the 29 July flight. Direct and diffuse sky irradiance can be highly variable along a given flight line, and can span clear-sky to white sky conditions with single and/or multi-layered cloud layers. In this near-polar geography and seasonal

period of snow and ice melt with expansive open water, low SZAs, and large energy fluxes between the surface and lower atmosphere result in dynamically changing measurement conditions over relatively short spatiotemporal scales. Collecting at-sensor zenith spectral irradiance during flight enables characterization of sky conditions to screen out unusable data related to cloud contamination as well as additional measurement information to support atmospheric compensation modeling. During the 29 July flight, the at-sensor VSWIR spectral irradiance calibration strategy was iteratively optimized for VNIR and

SWIR1/2 detectors to avoid irradiance saturation when flying above, in-between, and below cloud layers.

During instrument integration into the LaRC UC-12B aircraft, it became evident that the zenith OrangeCan design on the top of the aircraft would exclude the direct component of at-sensor VSWIR spectral irradiance at low illumination angles. During the 29 July flight, it was verified that the RCR optic did not receive the direct component of at-sensor VSWIR spectral

irradiance as would be the case at incident SZAs during the entire airborne mission. Based on this spectral irradiance measurement limitation, we removed the OrangeCan from the top of the LaRC UC-12B aircraft on the Thule Air Base (AB) tarmac once the aircraft returned from its daily flight line. Removing the OrangeCan from the top of the aircraft enabled the flight team to quantify its impact on direct and diffuse at-sensor VSWIR spectral irradiance measurements. This problem is addressed in Sections 3.3.2 and 3.3.3.

In addition to the OrangeCan's impact on in-flight measured at-sensor VSWIR spectral irradiance, we note another observational caveat that is tied to the imperfect cosine response of the RCR. Horizontal positional change of the LaRC UC-12B resulting from atmospheric turbulence and/or pitch, yaw, and roll maneuvers would result in a hemispherical spectral irradiance measurement bias, especially for the direct path. Under clear-sky or white sky conditions, it may be possible to

assess how horizontal changes in the LaRC UC-12 influenced in-flight spectral irradiance measurements in a post-processing mode. We deemed this to be negligible relative to the at-sensor VSWIR spectral irradiance measurement requirement, because the direct path was excluded, and the zenith position of the OrangeCan was only intended as a point of reference for sky conditions during flight.

### 3.3 Post-Flight Laboratory and Field Calibration Procedures

### 3.3.1 Nadir Viewing VSWIR Spectrometer IFOV Characterization

A NIST traceable source and integrating sphere in the NASA/GSFC Code 618 RCL clean room was used to measure the nadir viewing VSWIR spectrometer 1° foreoptic point spread function (PSF). A sliding optical rail with mm increments was mounted



on a laboratory table parallel to the integrating sphere aperture. The 1° foreoptic was mounted and aligned on the sliding optical rail at a distance of 101.5 cm from the 1° aperture to the integrating sphere aperture. Sliding from left to right in parallel (i.e., equivalent to cross-track vignetting (Chrien et al., 1990)) to the integrating sphere aperture, radiance measurements were captured in 1 mm increments. The measurement technique involved starting in an occulted left position, sliding the 1° aperture

across the integrating sphere output to measure the width of the 1° radiance response, and then finishing in an occulted right position (Figure 5). Using Eq.1, PSF in-IFOV and near-IFOV scale factors (sf) can be computed:

$$[\text{in-IFOV}_{PSFsf}, \text{near-IFOV}_{PSFsf}] = 1° \text{ aperture}_{width} - \text{integrating sphere aperture}_{width} , \qquad (1)$$

where the in-IFOVPSFsf excludes left and right edge aperture measurements (to the nearest mm), and near-IFOVPSFsf includes left and right edge aperture measurements (to the nearest mm). The 1° aperture width excluding edges was measured at 26.5 cm, and the 1° aperture width including edges was measured 26.9 cm. The integrating sphere aperture width is 25 cm. Using the in-IFOVPSFsf = 1.5 cm and near-IFOVPSFsf = 1.9 cm, the ground sampling footprint for at-sensor spectral radiance can be approximated with the Eq. 2:

$$IFOV_{ground} = \text{in-IFOV}_{PSFsf} \text{ or near-IFOV}_{PSFsf} \cdot \text{SIMPL Altitude}_{AGL} , \qquad (2)$$

where IFOVground is in meters, in-IFOVPSFsf or near-IFOVPSFsf is in meters (converted from cm), and SIMPL AltitudeAGL is the distance from the sensor to the surface in meters.

**3.3.2 Zenith Viewing VSWIR Spectrometer RCR Characterization**

The hemispherical irradiance response for the RCR optic was measured in the NASA/GSFC Code 618 RCL clean room using a 1000-Watt NIST traceable point source in dark conditions. Reflective stray light from any surface other than the point source in the clean room was blocked off with additional dark materials. The point source was mounted on a laboratory table directly behind a rectangular shaped bevel to constrain illumination rays. The RCR optic was secured to a rotating mount with an

angular resolution of 1°. Point source irradiance measurements were captured with the RCR optic placed inside the OrangeCan with the BK7 optical window as well as without the OrangeCan. This procedure was intended to repeat spectral irradiance measurements collected during the airborne mission, and to quantify the OrangeCan's impact on hemispherical irradiance measurements in a controlled laboratory environment.

Point source irradiance measurements for the RCR optic without OrangeCan obstruction were captured in 5° angular increments from 0° to 180°. OrangeCan RCR measurements were captured in 1° angular increments from 0° to 180°. The OrangeCan's impact on the RCR response is shown in Figure 6. We determined that the IFOV of the OrangeCan RCR optic mounted in a zenith position on top of the aircraft was 102° (to the nearest degree). Thus, for SZAs lower than 51°, the direct



component of at-sensor VSWIR spectral irradiance was not received by the zenith viewing VSWIR spectrometer RCR optic during the airborne mission.

### 3.3.3 RCR Field Experiment

The objective of the RCR field experiment was to determine how the spectral irradiance measurements collected in a zenith
position with the OrangeCan's 102° FOV could be useful for discriminating sky conditions during each flight. On 15 December
2015, we conducted a RCR verification experiment on the roof of Building 33 at NASA GSFC. The exact roof location was
adjacent to the AERONET calibration site (aeronet.gsfc.nasa.gov, 38.99250°N, 76.83983W), and provided an unobstructed
hemispherical IFOV. We used both non-imaging VSWIR spectrometers deployed during the airborne mission to coincidentally
collect hemispherical-sky and OrangeCan-sky RCR measurements mounted a level-tripods side by side at a temporal sampling
frequency of one second.

Given the known limitation that the OrangeCan RCR optic could not receive the direct component of at-sensor VSWIR spectral
irradiance at SZAs lower than 51°, we wanted to mimic the solar illumination geometry and both direct and diffuse-sky
conditions under plausible measurement scenarios during the airborne mission. Thus, four hemispherical-sky illumination
scenarios were evaluated: (1) direct clear-sky and diffuse clear-sky; (2) direct clear-sky and diffuse cloud-sky; (3) direct cloud-
sky and diffuse clear-sky; and (4) direct cloud-sky and diffuse cloud-sky. Both hemispherical-sky and OrangeCan-sky RCR
measurements were collected during the temporal window of 9am to 3pm (Eastern Standard local time). We monitored variable
solar illumination conditions and periodically photographed direct and diffuse-sky scenes to complement RCR measurements.
We selected hemispherical-sky and OrangeCan-sky RCR measurements for each illumination scenario described above. The
raw counts were converted to spectral irradiance using calibration coefficients and a parabolic correction. The coincident
(within one minute) hemispherical-sky and OrangeCan-sky RCR measurements accompanying each photographed scenario
were summarized using averaging.

Our hemispherical-sky/OrangeCan-sky RCR comparison shown in Figure 7 indicates that the OrangeCan-sky spectral
irradiance measurements from the airborne mission can be exploited to discriminate diffuse-sky conditions only. Our analysis
of sky condition scenarios indicates that when clouds are passing above the zenith mounted OrangeCan; the RCR spectral
irradiance response increases appreciably when compared to the diffuse clear-sky response. Our interpretation of this spectral
irradiance response is that clouds are diffusing light directly above (whether on ground or in-flight) where photons undergo
multiple scattering within and between single and/or multi-layered cloud strata. Despite the missing direct component of
hemispherical spectral irradiance, the diffuse OrangeCan-sky response serves as an important discriminator for sky conditions
more broadly during each flight (Figure 8). At a minimum, zenith measured sky conditions from the zenith viewing VSWIR



spectrometer during flight can inform appropriate selection of useable science quality airborne measurements from the nadir viewing VSWIR spectrometer.

## 4. Airborne VSWIR Spectrometer Measurement Performance

### 4.1 Benchmarking Methodology

The preferred practice in airborne VSWIR measurement science is to have a complementary ground cal/val experiment designed around acquiring ground reflectance of a known pseudo-invariant target (e.g., desert playas) during overflight with a field spectrometer (Green et al., 1993;Green et al., 1998;Slater et al., 1987;Thome, 2001;Thompson et al., 2015). Under this scenario, a hemispherical calibrated Spectralon reference panel is used to characterize incident downwelling irradiance to enable the derivation of a remote sensing reflectance spectrum. This remote sensing reflectance spectrum combined with real

time atmospheric measurements, namely aerosol properties and columnar water vapor, are used to constrain an atmospheric radiative transfer model for calculating at-sensor spectral radiance for the airborne instrument. Another technique is to model apparent airborne surface reflectance using radiative transfer, and then re-scale to ground reflectance using an empirical line correction (Gao et al., 1993;Moran et al., 2001;Smith and Milton, 1999).

For the SIMPL/AVIRIS-NG 2015 Greenland campaign, no ground or ship campaign occurred over the Greenland Ice Sheet or sea ice, which were the primary measurement targets of interest. Logistical challenges and cost prevented ground deployment on the Greenland Ice Sheet or ship deployment on the open ocean for purposes of acquiring in situ measurements. However, on 14 August 2015, a small scale cal/val experiment was conducted on a tarmac at Thule AB where both NASA LaRC and Dynamic Aviation aircraft carrying the airborne non-imaging VSWIR spectrometers and AVIRIS-NG flew near

simultaneously acquiring measurements over dark asphalt. Our initial focus in this paper is to document the radiometric calibration methods for deployment of the non-imaging airborne VSWIR spectrometers, and to assess the nadir viewing VSWIR spectrometer's measurement performance over dark and bright Greenland targets during the in-flight radiometric calibration experiment. We plan to benchmark the nadir viewing VSWIR spectrometer's measurement performance against AVIRIS-NG for the Thule AB cal/val experiment, but reserve that effort for future investigation.

Given our ground campaign constraints, we developed an alternative method to benchmark measurement performance based on MODTRAN along with a coincident Landsat 8 OLI image acquisition comparison. This alternative method involved selecting two independent flight line segments over homogenous bright Greenland ice and dark bare rock/soil targets using both high resolution camera images, and the 29 July Landsat 8 OLI image (Figure 9). As an additional check for these dark

and bright target segments, we used the zenith viewing VSWIR spectral irradiance measurements to confirm that variance in measured at-sensor spectral radiance was not related to changing solar illumination conditions during flight. For these dark and bright target segments, we converted the average at-sensor spectral radiance measured with the nadir viewing VSWIR



spectrometer to apparent reflectance using Eq. 3. The resulting reflectance spectra for bright Greenland ice and dark bare rock/soil targets are shown in Figure 10.

$$\rho_\lambda = (\pi L_\lambda \, d^2) / (Eo_\lambda \cos(\Theta)) \tag{3}$$

where

$\rho$ = apparent reflectance
L = VSWIR radiance
d = sun-earth distance in au
10      Eo = TOA irradiance (Kurucz, 2005)
$\Theta$ = solar zenith angle

To increase accuracy of the calculations, knowledge about the surface reflectance is required to partition light scattering and absorption within the spectrometer's LOS.

## 4.2 Airborne At-Sensor Simulation with MODTRAN

Water vapor and aerosols are the two most significant attenuation factors effecting downward and upward atmospheric transmittance of spectral radiance along the direct path and LOS. The nadir viewing VSWIR radiances were benchmarked against MODTRAN6 (Berk et al., 2017) calculated at-sensor spectral radiances for both the bright and dark targets. Simulating at-sensor VSWIR spectral radiance for bright and dark targets along the 29 July flight line, required atmospheric aerosol and 20 columnar water vapor measurements from a variety of sources. The northwestern portion of the Greenland Ice Sheet is quite remote with sparse ground instrumentation to parameterize MODTRAN, especially towards the Greenland interior. On the coast at the Thule AB, there is an AERONET site with a CIMEL maintained by NASA GSFC. The CIMEL measurements provided spectral AOD, aerosol extinction coefficients, and columnar water vapor, as the source of atmospheric information. We also used carbon dioxide and water vapor measurements from the Atmospheric Infrared Sounder (AIRS) and MODerate 25 resolution Imaging Spectrometer (MODIS) Terra and Aqua instruments.

MODTRAN has four core model components [i.e., (1) radiation transport; (2) aerosol and clouds; (3) LOS geometry; and (4) spectral range and resolution] that are required for simulating atmospheric conditions while the airborne spectrometer is in-30 flight (Berk et al., 2016). The following options were selected: the sub-arctic summer model atmosphere; correlated-k algorithm to prescribe radiation transport at a spectral resolution of 0.1 cm$^{-1}$; the Kurucz 2005 TOA solar irradiance reference spectrum (Kurucz, 2005); the Navy maritime aerosol model weighted for stronger coastal than continental influence; and meteorological range based on the CIMEL-retrieved aerosol extinction coefficient at 550 nm. Other parameters included ozone and carbon dioxide concentrations along with columnar water vapor content (g/cm$^{-2}$) from atmospheric measurements on 29 35 July described above.



The LOS geometry was determined using the NASA LaRC aircraft flight altitude (based on the navigation file), an observer zenith angle of $180^0$, and the ground altitude was extracted from the Greenland Ice Mapping Project (GIMP) Digital Elevation Model (Howat et al., 2014). The Julian day and in-flight start time for data acquisition was used to initialize the solar illumination geometry parameters that included observer latitude and solar zenith angle. Finally, we convolved MODTRAN output radiances into VSWIR channels using a Gaussian FWHM filter centered on 1 nm wavelengths from 0.35 to 2.5 μm. The spectral response functions for the nadir viewing VSWIR spectrometer VNIR and SWIR detectors are shown in Figure 11.

### 4.2.1 Dark and Bright Target At-Sensor Simulations

MODTRAN assumes the atmosphere to be horizontally homogeneous – at some point the assumption starts to break down. With regard to water vapor, we can quantify that breaking point with the geodetic distance from the Thule AB CIMEL to the dark and bright targets. Each target presented a different set of challenges in the benchmarking process. Along Greenland's ice margin, glacial moraines and bedrock are comprised of rock and soil mixtures often lacking surface homogeneity. Fortunately, the dark target location is only 54.22 km from the Thule AB CIMEL. The water vapor and aerosols retrievals coincident to the time of the VSWIR acquisition were used to successfully constrain MODTRAN (Figure 12). However, the atmospheric conditions prevailing over the bright land ice target were even more challenging to parameterize due to the geodetic distance of 150.35 km from the Thule AB CIMEL. While the CIMEL-retrieved aerosol loadings appeared to be indicative of the land ice target, the water vapor was not. Additionally, for satellite image data, it can be difficult to partition aerosol scattering from bright snow and ice surface scattering because atmospheric aerosols have relatively low reflectance by comparison (Istomina et al., 2011), and therefore, we did not attempt to use satellite aerosol retrievals.

We did not consider applying a nonlinear least squares spectral fitting algorithm of the water vapor absorption features of the VSWIR bright land ice radiance spectra as we are in the process of validating the non-imaging airborne spectrometer; instead, we chose well calibrated satellite sensor retrievals for a scientific, transparent approach. Water vapor is an initial atmospheric condition that can be spatially variable across coastal to inland gradients, particularly during the Greenland summertime melt period when surface to atmosphere latent heat fluxes are strong. Thus, we opted to exploit a range of water vapor measurements (Table 1) over the Greenland interior to constrain MODTRAN's sensitivities to critical absorption features (Figure 13). At 67° N, the spatial footprint of the 1° x 1° gridded daily MODIS L3 Aqua water vapor product (Platnick et al., 2015) is approximately 44 km spatial resolution. The "low mean" appeared to best fit our data.

### 4.2.2 Landsat 8 OLI At-Sensor Simulation with MODTRAN

As described earlier in the paper, Landsat 8 OLI's orbital tracks converge towards the poles, and for northwestern Greenland, that results in considerable imaging swath side lap during the sunlit summer season. On 29 July, a coincident image for World Reference System-Two (WRS-2) Path 26 Row 05 was acquired over the Greenland Ice Sheet interior during the NASA LaRC



flight. We identified the overlapping region where the bright land ice target flight segment intersected with the Landsat 8 OLI Collection One image data (available at https://earthexplorer.usgs.gov/). Using the NASA LaRC Applanix data and aircraft navigation information, we identified the closet Landsat 8 OLI pixels that corresponded to the nadir viewing VSWIR spectra along the bright land ice flight segment. Using the bright land ice MODTRAN prescription for the nadir viewing VSWIR

spectrometer, we simulated at-sensor TOA spectral radiance for Landsat 8 OLI using solar illumination geometry, swath LOS imaging geometry, relative spectral response functions, and the apparent bright land ice remote sensing reflectance spectra. There was no discernible cloud contamination for Landsat 8 OLI pixels. We rescaled Landsat 8 OLI at-sensor digital counts to at-sensor TOA spectral radiance using radiance-based calibration coefficients contained within the image metadata. Finally, we compared MODTRAN simulated Landsat 8 OLI at-sensor TOA spectral radiances for the bright land ice target with

observed Landsat 8 OLI at-sensor TOA spectral radiances. The comparison was based on the average radiance from 24 nadir viewing VSWIR spectra, and 24 Landsat 8 OLI pixels.

## 5. Results and Discussion

A method to radiometrically calibrate, deploy and benchmark measurement performance of a non-imaging airborne VSWIR spectrometer to measure the Greenland Ice Sheet surface has been presented. This NIST traceable calibration included rigorous

laboratory, in-flight, and field procedures to fully characterize VSWIR spectrometers, their foreoptics, and their at-sensor measurements. The nadir viewing VSWIR spectrometer's stability was determined to be less than 2% using a NIST traceable source, and well within the targeted 5% at-sensor VSWIR spectral radiance accuracy requirement for the airborne mission. The point spread function and IFOV footprint of the nadir viewing VSWIR spectrometer's 1° foreoptic was measured to enable direct comparison to SIMPL's green and NIR polarimetric lidar measurements, AVIRIS-NG's VSWIR measurements, and

other on-orbit satellite measurements such as Landsat for example. The 29 July in-flight radiometric calibration experiment over Greenland bright and dark targets proved to be invaluable for optimizing the nadir viewing VSWIR spectrometer's measurement capabilities during the airborne mission, as well as evaluating in-flight measurement performance across the full solar spectrum dynamic range using MODTRAN and atmospheric measurements from both ground and satellite instruments. The main objective of measuring at-sensor VSWIR spectral irradiance with a zenith viewing VSWIR spectrometer and RCR

foreoptic, was to characterize in-flight sky conditions. Even though the zenith mounted OrangeCan on top of the NASA LaRC aircraft limited the hemispherical IFOV, these measurements are useful for screening out unusable flight data that will expedite identification of science quality VSWIR data the can be used to address airborne mission objectives.

With no ground cal/val in situ measurements on the Greenland Ice Sheet, or ship campaign on open ocean, we had to develop

an alternative approach to benchmark the nadir viewing VSWIR spectrometer's measurement performance using an atmospheric radiative transfer model. By identifying homogenous bright land ice and dark bare rock/soil flight segments on 29 July, we were able to assess airborne measurement performance with MODTRAN over both low and high radiance targets (e.g., (Moran et al., 1995)) under very similar atmospheric and solar illumination conditions. We used apparent airborne remote





sensing reflectance spectra for both bright and dark targets to simulate at-sensor VSWIR spectral radiance for the nadir viewing
VSWIR spectrometer, and then compared simulations with observed at-sensor VSWIR spectral radiance (e.g. (Green,
2001;Slater et al., 1987;Thome, 2001;Vane et al., 1993). Our MODTRAN simulations indicate that the nadir viewing VSWIR
spectrometer VNIR and SWIR1 detectors measured bright land ice on average between 2.5 – 4.7% accuracy for VSWIR

wavelengths with greater than 80% atmospheric transmittance (Figure 14). For dark bare rock/soil, the nadir viewing VSWIR
spectrometer VNIR and SWIR1 detectors measured between 0.6 – 1.2 % accuracy on average (Figure 14). As stated earlier,
NASA LaRC optical window transmission beyond 2.0 µm was more uncertain and was evident when evaluating the SWIR2
detector data. For bright land ice and dark bare rock/soil, the nadir viewing VSWIR spectrometer's measurement accuracy for
the SWIR2 detector was on average 4.3% and 19.7%, respectively (Figure 14).

The accuracy of MODTRAN at-sensor calculations for benchmarking airborne remote sensing instrument performance is in
part, dependent on the quality of the surface reflectance spectra and availability of atmospheric measurements near the target
measurement performance location. Fortunately, for this airborne campaign, baseline atmospheric measurements were
accessible via the Thule AB CIMEL as part of AERONET. It is clear that spatial proximity to a CIMEL matters in terms of

in-flight atmospheric aerosols and columnar water vapor concentrations because we were able to achieve greater VSWIR
measurement accuracy for the closer dark bare rock/soil target when compared to the bright land ice target much farther way.
Interestingly, we found that the nadir viewing VSWIR spectra for bright land ice in the Greenland interior was much more
sensitive to columnar water vapor concentrations than aerosols. This result caused us to evaluate the nadir viewing VSWIR
spectra sensitives to a variety of input satellite atmospheric water vapor products. Narrowing in on 0.94 µm and 1.13 µm water

vapor absorption lines uncovered the spread in satellite retrieved daily atmospheric water vapor over the Greenland interior.
We were able to identify that the MODIS Aqua Low Mean atmospheric water vapor product is most suitable to ingest when
processing the NASA LaRC science flight data for MODTRAN-based atmospheric compensation. The daily MODIS Aqua
overpass times generally align well with NASA LaRC flight times during the airborne mission. The MODIS Aqua Low Mean
atmospheric water vapor retrievals are designed to partition columnar water vapor concentrations between the surface and 680

mb (see details at https://modis-atmosphere.gsfc.nasa.gov/documentation/collection-6.1), which is within the atmosphere
boundary layer.

As an additional airborne VSWIR spectrometer performance comparison over the Greenland Ice Sheet, we used a Landsat 8
OLI coincident image acquired within ~ 3 minutes of the NASA LaRC bright land ice target flight segment. We simulated

Landsat 8 OLI at-sensor TOA spectral radiance using MODTRAN using the following constraints: solar illumination
geometry, OLI viewing geometry, the same atmospheric input parameters used for the airborne VSWIR spectrometer
assessment, and the apparent airborne remote sensing reflectance spectrum for bright land ice. By comparing MODTRAN
simulated and observed Landsat 8 OLI at-sensor TOA spectral radiance, we found that Landsat 8 OLI is measuring between 6
and 16% more at-sensor TOA spectral radiance from the Greenland Ice Sheet with VNIR and SWIR1 spectral bands than was





simulated with the nadir viewing VSWIR spectrometer's apparent airborne remote sensing reflectance spectrum (Figure 15). It is important to note that Landsat 8 OLI's pixel-level LOS imaging is highly accurate over Greenland due to spacecraft geolocation (Storey et al., 2014), and that we accounted for cross-track imaging effects in MODTRAN using NIR spectral band LOS geometry.

Landsat 8 OLI is a well characterized instrument on both pre- and post-launch timescales with exceptional on-orbit performance since 2013 (Markham et al., 2014;Morfitt et al., 2015). Routine on-board diffuser, lunar, and vicarious calibrations over mid-latitude pseudo invariant calibration sites (PICS) in particular, are conducted to track OLI's instrument performance and degradation while in orbit (Helder et al., 2013;Helder et al., 2010;Mishra et al., 2014). We speculate that

differences between simulated and observed Landsat 8 OLI at-sensor TOA spectral radiance over the Greenland Ice Sheet presented in this paper, are possibly a by-product of both techniques used to derive OLI gain coefficients over mid-latitude desert sites with stable dry atmospheres, and VNIR differences between the Kurcuz and ChKur reference TOA solar irradiance spectrums (Chance and Spurr, 1997;Kurucz, 2005) used for airborne VSWIR spectrometer and Landsat 8 OLI radiometric cal/val. Nevertheless, more investigation is required and looking ahead, we recommend that Greenland and Antarctica ice

sheets receive expanded cal/val consideration when characterizing and monitoring on-orbit satellite instrument performance, as has been attempted for other Earth observing systems (Cao et al., 2010;Six et al., 2004). The VSWIR airborne method of cal/val presented here, including the rigorous NIST traceable radiometric calibration, is put forth as an option to augment polar ice sheet cal/val.

It has been suggested that optical remote sensing instruments must be able to measure the ice sheet surface at an accuracy of 2% or less to distinguish between the presence of light absorbing constituents and other factors controlling VSWIR ice sheet albedo (Warren, 2013). For airborne and on-orbit satellite instruments, this stringent of a measurement requirement demands careful instrument radiometric calibration and characterization and could remain difficult to achieve for polar atmospheres because of atmospheric measurement uncertainty and the ability to compensate for such effects. Landsat 8 OLI's capabilities

to measure Greenland and Antarctica ice sheets has advanced since 2013 thanks to revisions in its higher latitude and polar image frequency (Fahnestock et al., 2016). While Landsat 8 OLI measurements are providing new insights and applications for polar ice sheet science, specifically supraglacial lake and ice velocity mapping (Alley et al., 2018;Gardner et al., 2018;Pope et al., 2016), results from this study suggest that the Greenland Ice Sheet surface may be less reflective than what is currently being measured by Landsat 8 OLI at TOA. Thus, Landsat 8 OLI reflectance-based interpretations of ice sheet surface properties

and change should remain cautious until additional measurement validation is undertaken. This initial effort to describe and document the radiometric calibration and measurement performance of the non-imaging airborne VSWIR spectrometer configuration flown as part of the SIMPL/AVIRIS-NG 2015 Greenland campaign, indicates that the nadir viewing VSWIR spectrometer was able to achieve its targeted at-sensor VSWIR measurement accuracy for the airborne mission as benchmarked against MODTRAN. Thus, we endorse and encourage the use of airborne VSWIR data products from NASA LaRC science





flights as they are of sufficient radiometric accuracy to evaluate green laser pulse penetration into Greenland snow and ice, and to evaluate other VSWIR remote sensing measurements acquired during the airborne mission timeframe.

## Funding Information

The ICESat-2 Project Science Office supported the SIMPL/AVIRIS-NG 2015 Greenland campaign, and Christopher Crawford's radiometric calibration work as part of a NASA Cooperative Agreement to the University of Maryland's Earth System Science Interdisciplinary Center. The MODTRAN and Landsat 8 components of this work were supported by a U.S. Geological Survey science support services contract to the Arctic Slope Regional Corporation (ASRC) Federal InuTeq as part of Christopher Crawford's USGS-NASA Landsat Science Team research.

## Acknowledgments

We would like to extend our grateful thanks for the generous contributions of the following people:  NASA GSFC Code 610 personnel for providing the VSWIR spectrometers, instrument calibration, and optics laboratory support resources; the SIMPL and AVIRIS-NG instrument teams and the pilots and ground crews of NASA LaRC and Dynamic Aviation; Brent Holben and the AERONET team at NASA GSFC for providing and processing the Thule AB CIMEL measurements; Rose Dominguez at NASA Ames Research Center for processing the NASA-LaRC Applanix flight data; Robert O. Green at the Jet Propulsion Laboratory for his recommendation to characterize the 1° foreoptic PSF for the nadir viewing VSWIR spectrometer.

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





**Figure 1**. The zenith mounted OrangeCan on top of the NASA LaRC UC-12B aircraft. The RCR optic was mounted inside directly under a BK7 optical window to characterize sky conditions during flight.





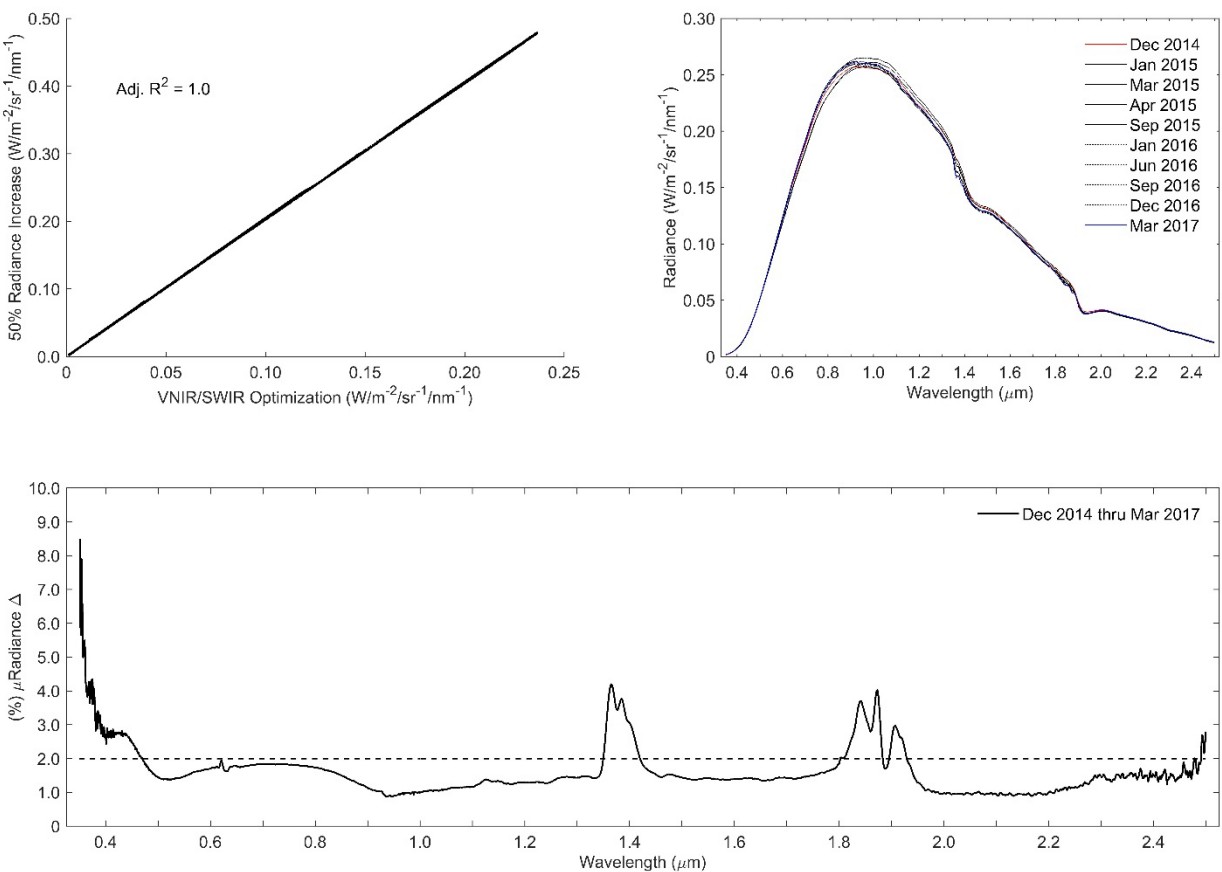

**Figure 2**. Laboratory calibration of the nadir viewing VSWIR spectrometer (FieldSpec Pro) using the NASA/GSFC Code 618 NIST traceable source. The top left panel scatterplot shows linearity test results using least-squared regression for 350-2500 nm. The top right panel show the integrating sphere radiance output (two lamp dark level) from the NIST traceable source during instrument repeatability checks. The line plot in the bottom panel summarizes FieldSpec Pro instrument stability by

5   wavelength over a ~2.5 year period. The dotted line signifies the achieved stability requirement.





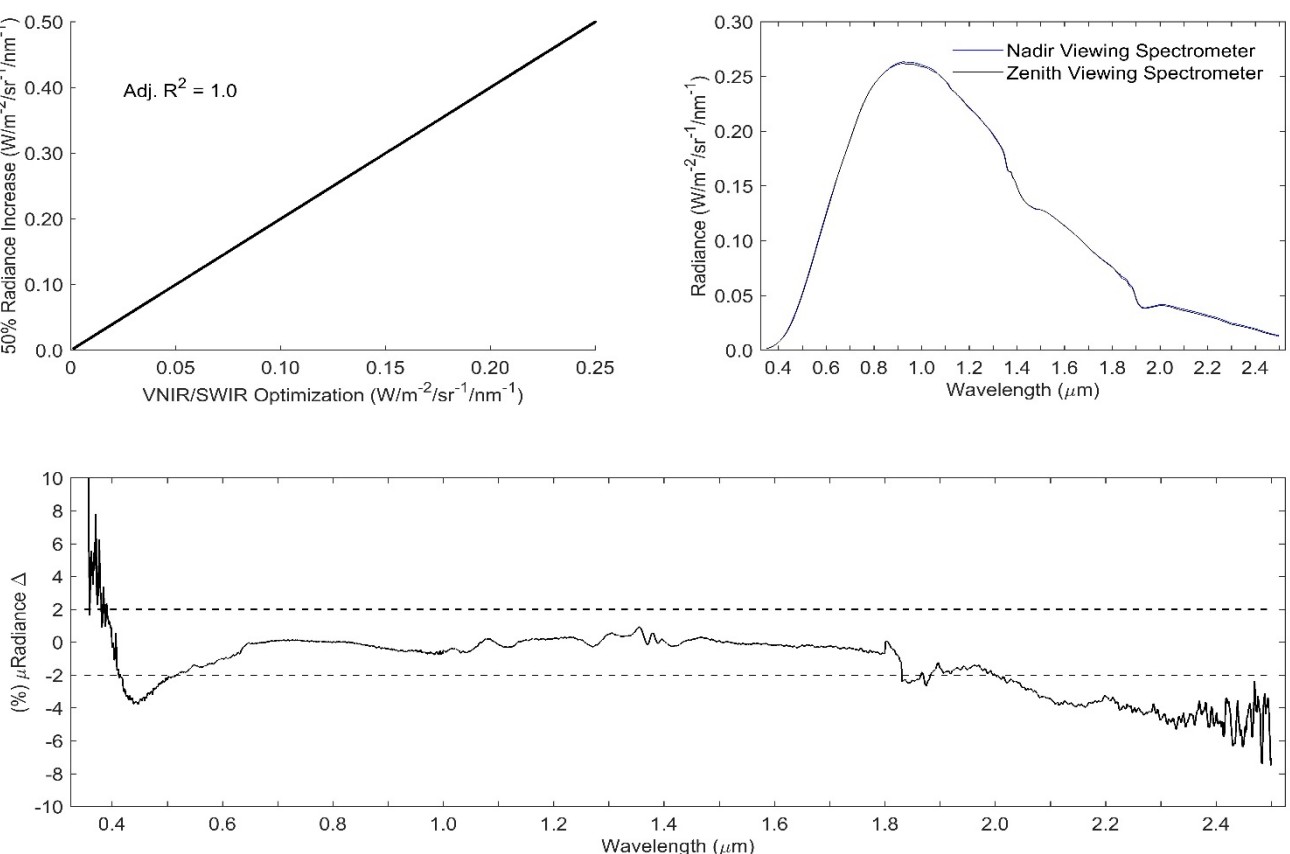

**Figure 3**. Laboratory cross-calibration of the nadir and zenith viewing (FieldSpec 3) VSWIR spectrometers using the NASA/GSFC Code 618 NIST traceable source. The top left panel shows the FieldSpec 3 linearity test results, and the top right panel shows the cross-calibration using the integrating sphere radiance output from the NIST traceable source. The line plot in the bottom panel summarizes the difference in response between nadir and zenith viewing VSWIR spectrometers relative to the achieved stability requirement (dotted lines).





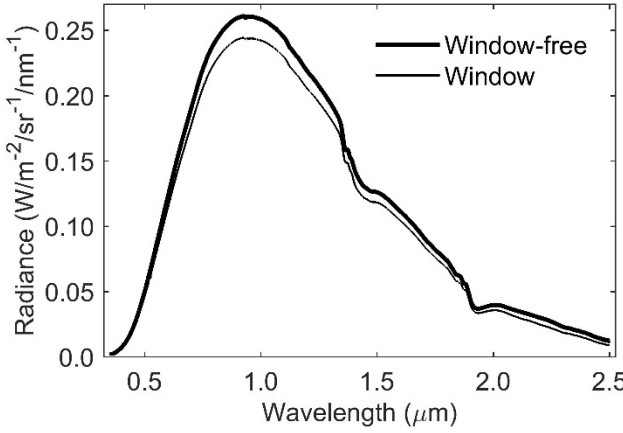 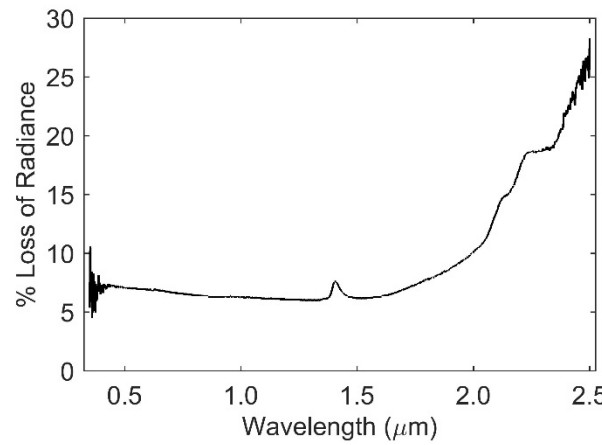

**Figure 4**. A measure of light transmission through the BK7 optical window mounted within the OrangeCan. The left panel shows the integrating sphere radiance output from the NIST traceable source with and without the optical window. The right panel summarizes wavelength-dependent radiance loss due to window transmissivity.





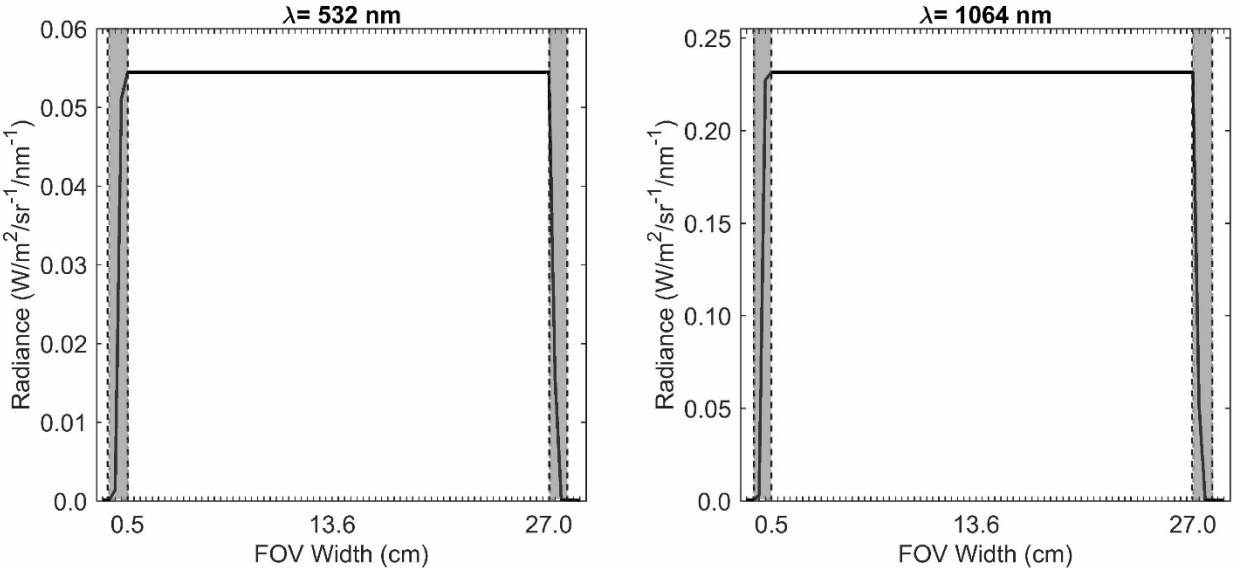

**Figure 5**. Laboratory characterization of the nadir viewing 1° foreoptic lens PSF and IFOV using the integrating sphere radiance output from a NIST traceable source. Results from green (left panel) and NIR (right panel) wavelengths at which SIMPL operates were used to summarize in-IFOV (thick black line within the dotted line boundaries) and near-IFOV widths (gray regions within the dotted lines).



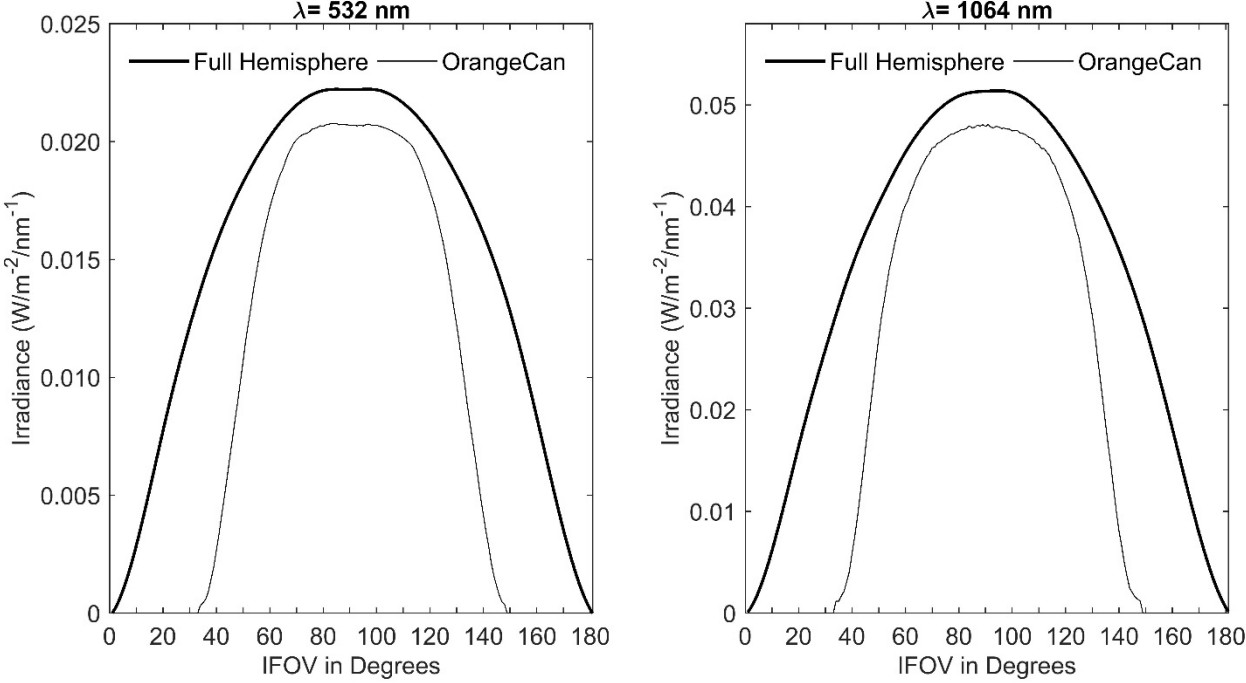

**Figure 6**. Laboratory characterization of the zenith viewing RCR optic using a NIST traceable point source. Green (left panel) and NIR (right panel) wavelengths at which SIMPL operates were used to summarize the OrangeCan's impact on the RCR optic IFOV and measured irradiance.





**Figure 7**. Remote cosine receptor field experiment results from 15 December 2015. Four separate solar illumination scenarios are represented with coincident hemispherical-sky and OrangeCan-sky at-sensor spectral irradiance measurements. Average spectral irradiance for each scenario was calculated using one-second measurement sampling for local time and SZA shown. Solar illumination conditions along the direct path and zenith diffuse-sky are shown on the right with photographs. Note: The amount of irradiance is dependent on the temporal proximity to solar noon, which on 15 December 2015 was 11:51 EST.

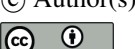



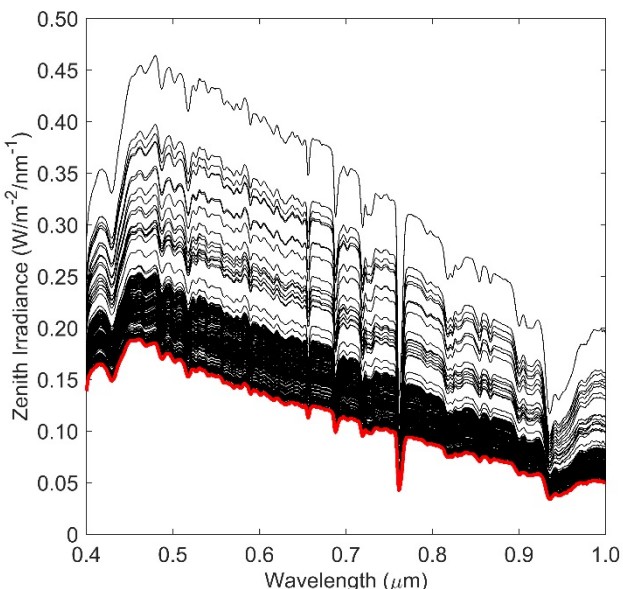
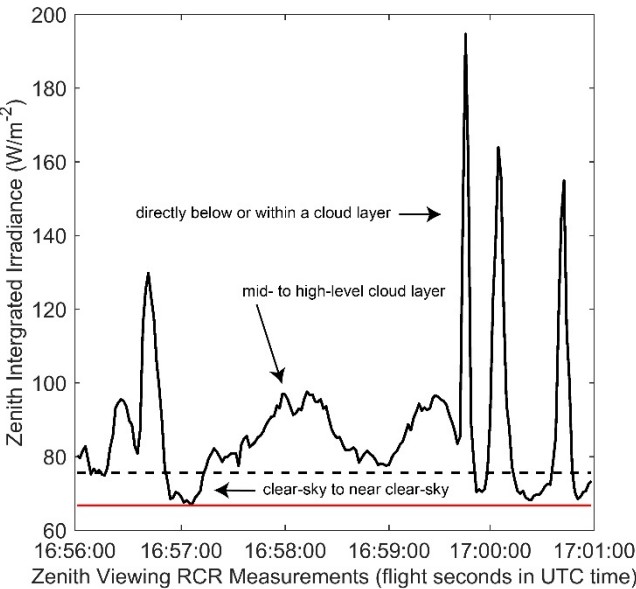

**Figure 8**. Example zenith RCR at-sensor irradiance measurements for a 29 July flight segment. The left panel shows zenith at-sensor irradiance measurements from 0.4 to 1.0 µm. The black lines indicate variability in instantaneous in-flight irradiance for a 5-minute flight segment. The thick red line signifies the baseline minimum irradiance received at-sensor, a condition that represents diffuse clear sky to near clear sky as verified with Figure 7 results. The right panel shows zenith integrated irradiance (i.e., sum function) from 0.4 to 1.0 µm for the same 5-minute flight segment. The thick black line indicates temporal variance in zenith integrated irradiance, a measure of sky conditions above the NASA LaRC aircraft. The dotted line signifies the computed mode (most frequently occurring condition) of zenith integrated irradiance, an indicator of sky condition stability. The red line serves as the minimum zenith integrated irradiance baseline. Using the temporal variance in zenith integrated irradiance, the mode value, and the minimum value, variable sky conditions during flight can be classified and the nadir viewing VSWIR spectrometer measurements can be filtered for data quality and scientific use.

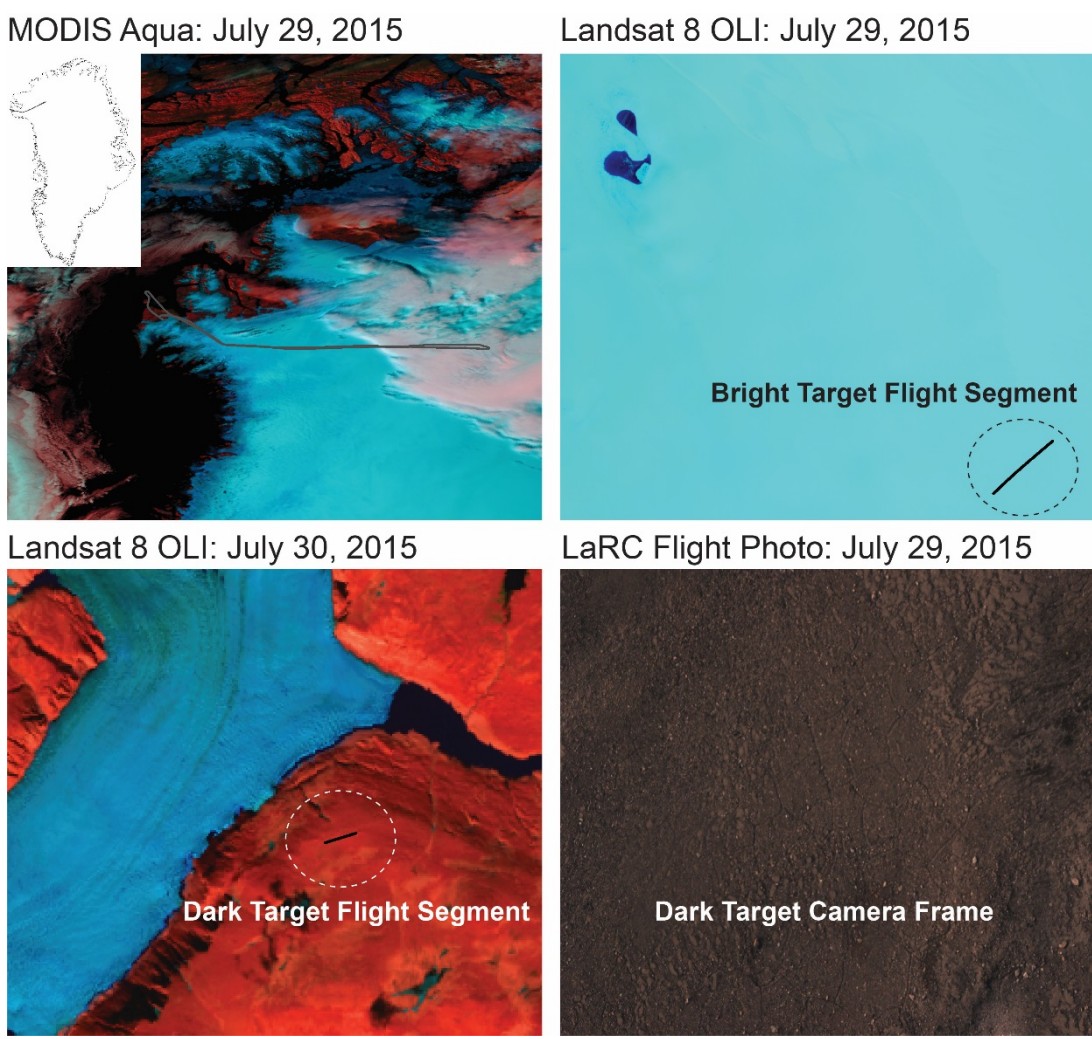

**Figure 9.** The 29 July flight line showing bright and dark target MODTRAN benchmarking segments for the VSWIR spectrometer. The upper left panel shows a MODIS Aqua image (false color SWIR, NIR, Green composite) with the NASA LaRC flight line (grey line). The upper right panel shows a Landsat 8 OLI image (false color SWIR, NIR, Green composite) with the bright land ice target flight segment (black line within the black dotted circle). The lower left panel shows a Landsat 8 OLI image (false color SWIR, NIR, Green composite) with the dark bare rock/soil target flight segment (black line within the white dotted circle). The lower right panel shows a NASA LaRC high resolution visible camera image (true color Red, Green, Blue composite) frame of the dark bare rock/soil target flight segment. Note, high resolution visible camera images were acquired over land ice during the campaign science flights.

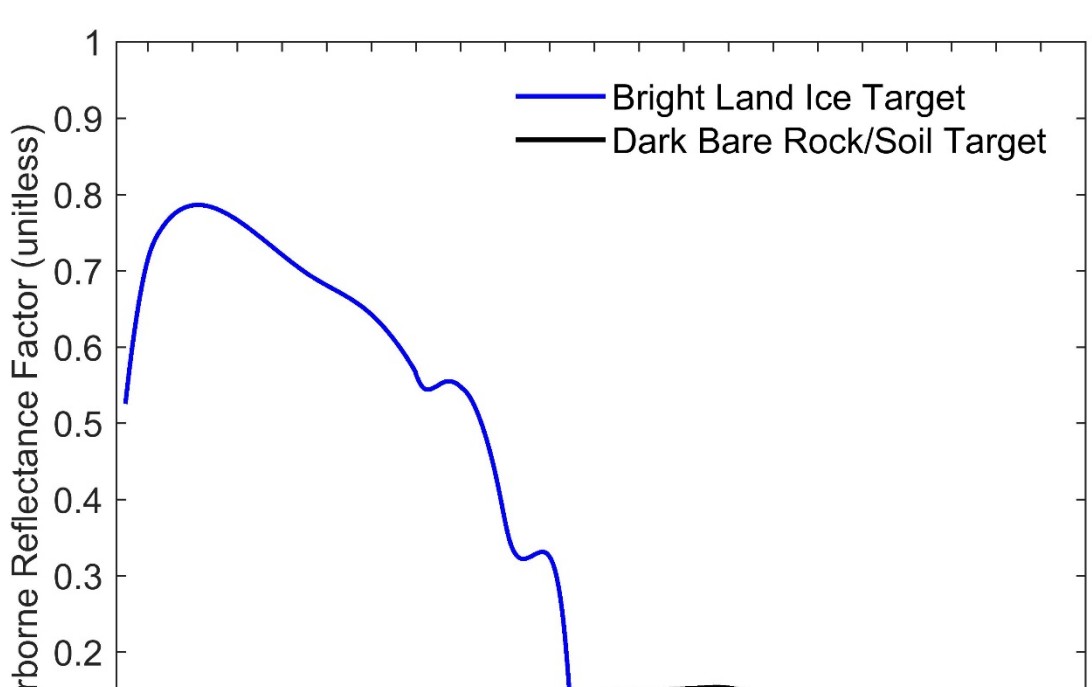

**Figure 10**. Apparent remote sensing reflectance spectra for bright and dark in-flight targets measured with the nadir viewing
VSWIR spectrometer.





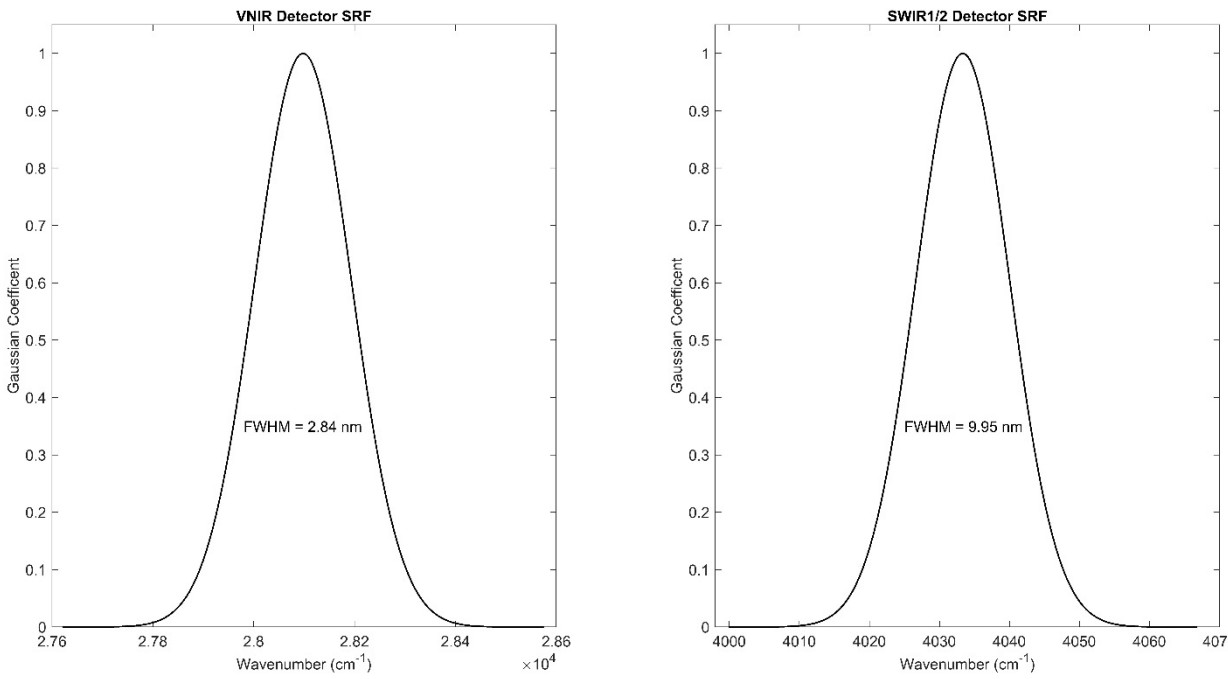

**Figure 11.** Gaussian spectral response functions for the airborne nadir viewing VSWIR spectrometer. The left panel shows the VNIR detector SRF, and the right panel shows the SWIR1/2 detector SRF. Note, FWHM refers to full width half maximum response to a filter value of 1.0 on the center wavelength.




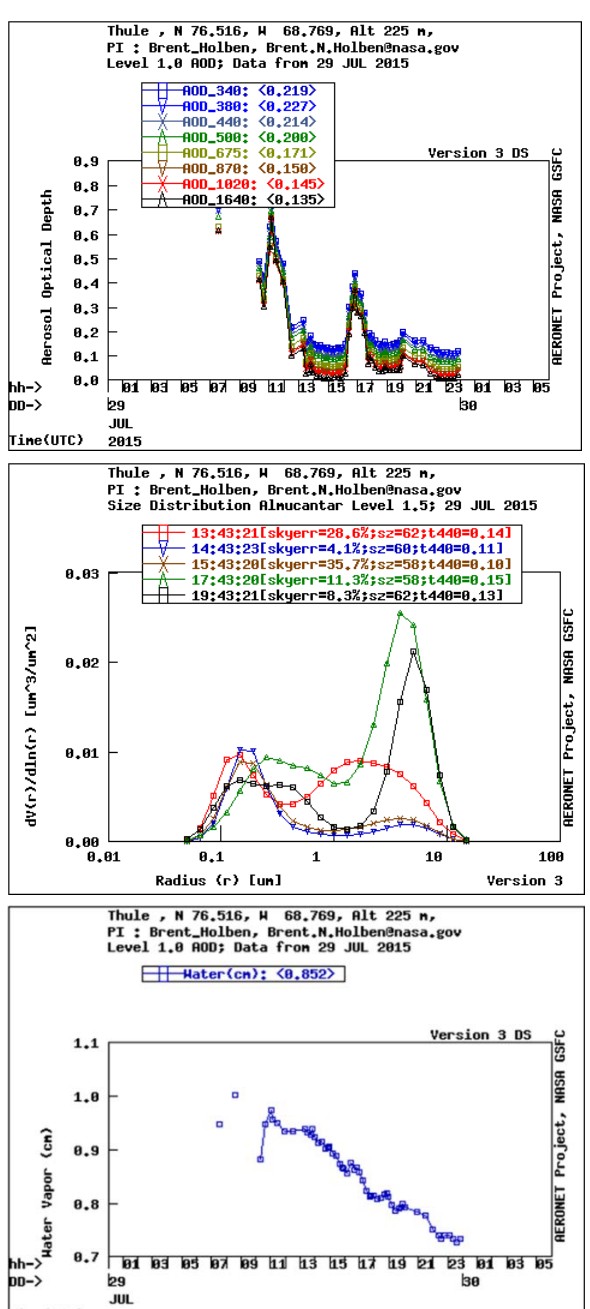

**Figure 12**. The CIMEL plots show the variability of the spectral AOD (top panel), aerosol size distribution (middle panel), and water vapor (bottom panel) throughout the day (29 July). The VSWIR acquisition time for the dark bare rock/soil target is 17.00 UTC (decimal time); bright land ice target 16.68 UTC. The meteorological range based on the $AOD_{550}$ is 67.27 km and 95.48 km, respectively.





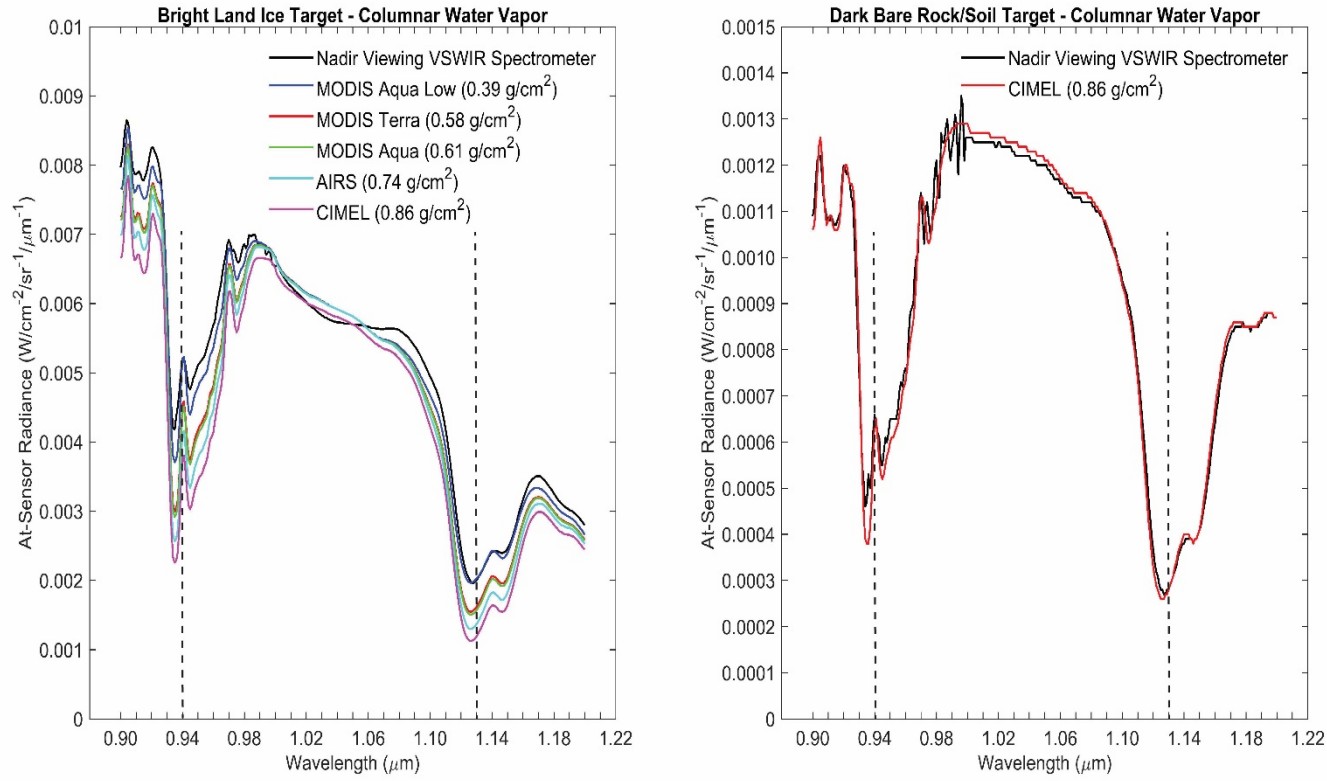

**Figure 13**. The airborne nadir viewing VSWIR spectrometer's at-sensor measurement sensitives to columnar water vapor for bright land ice and dark bare rock/soil targets. A variety of satellite columnar water vapor data products where evaluated for the bright land ice target due to the remoteness of the flight line segment and its proximity to the Thule AB CIMEL. Columnar water vapor absorption lines centered at 0.94 µm and 1.13 µm are designated with black dotted lines.





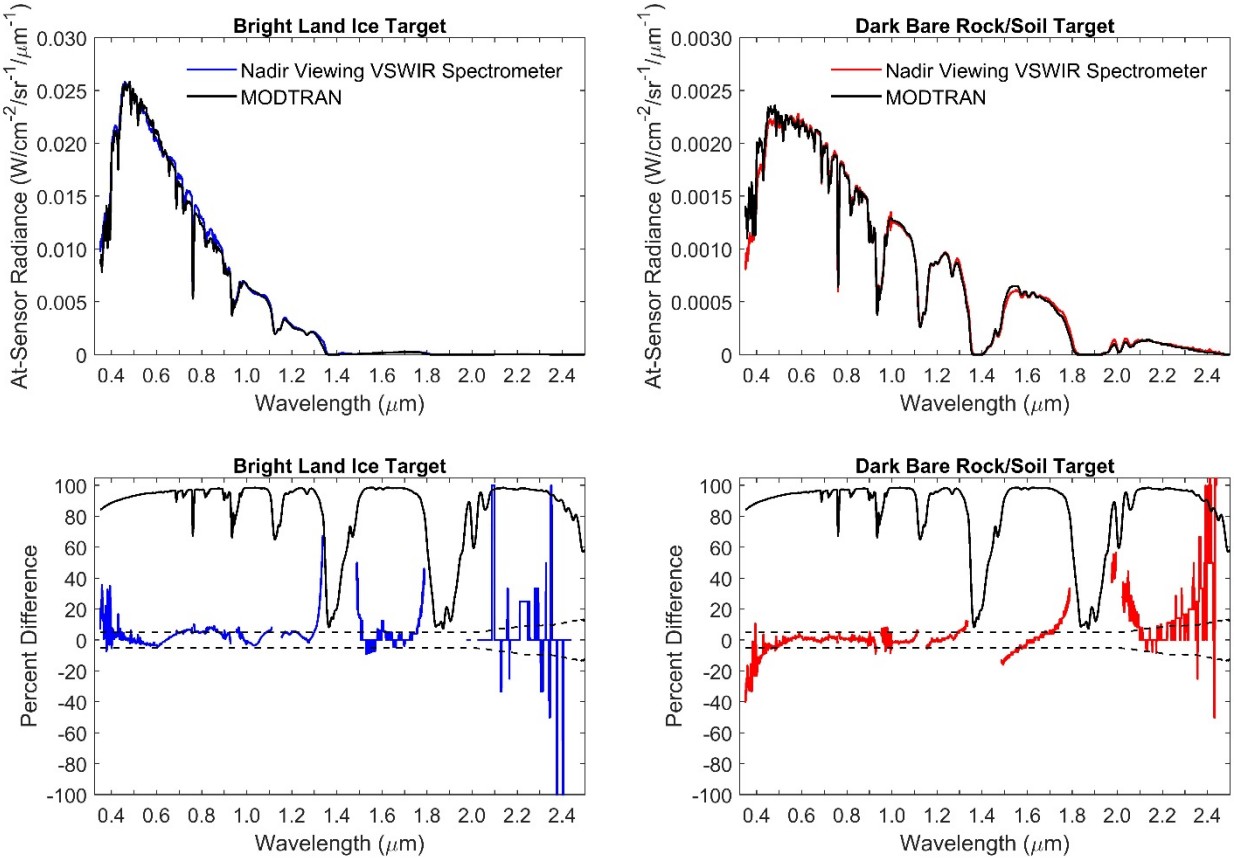

**Figure 14**. The airborne nadir viewing VSWIR spectrometer's at-sensor measurement performance for bright and dark targets as benchmarked against MODTRAN. The top left panel shows a comparison between simulated and observed at-sensor radiance for bright land ice. The top right panel shows a simulated verse observed at-sensor comparison for dark bare rock/soil. The bottom left panel describes the percent difference [i.e., percent difference= (observed – simulated) / simulated] between simulated and observed VSWIR spectrometer at-sensor radiance for bright land ice (blue line). The percent difference for the dark bare rock/soil target is shown in the bottom right panel. The dotted and top thick black lines on both panels signify the at-sensor measurement requirement and simulated atmospheric transmittance, respectively. The VSWIR spectrometer's measurement performance beyond 2.0 µm is subject to noise created by NASA LaRC BK-7 window transmission, and low to relatively low at-sensor SWIR radiances for both bright and dark targets.




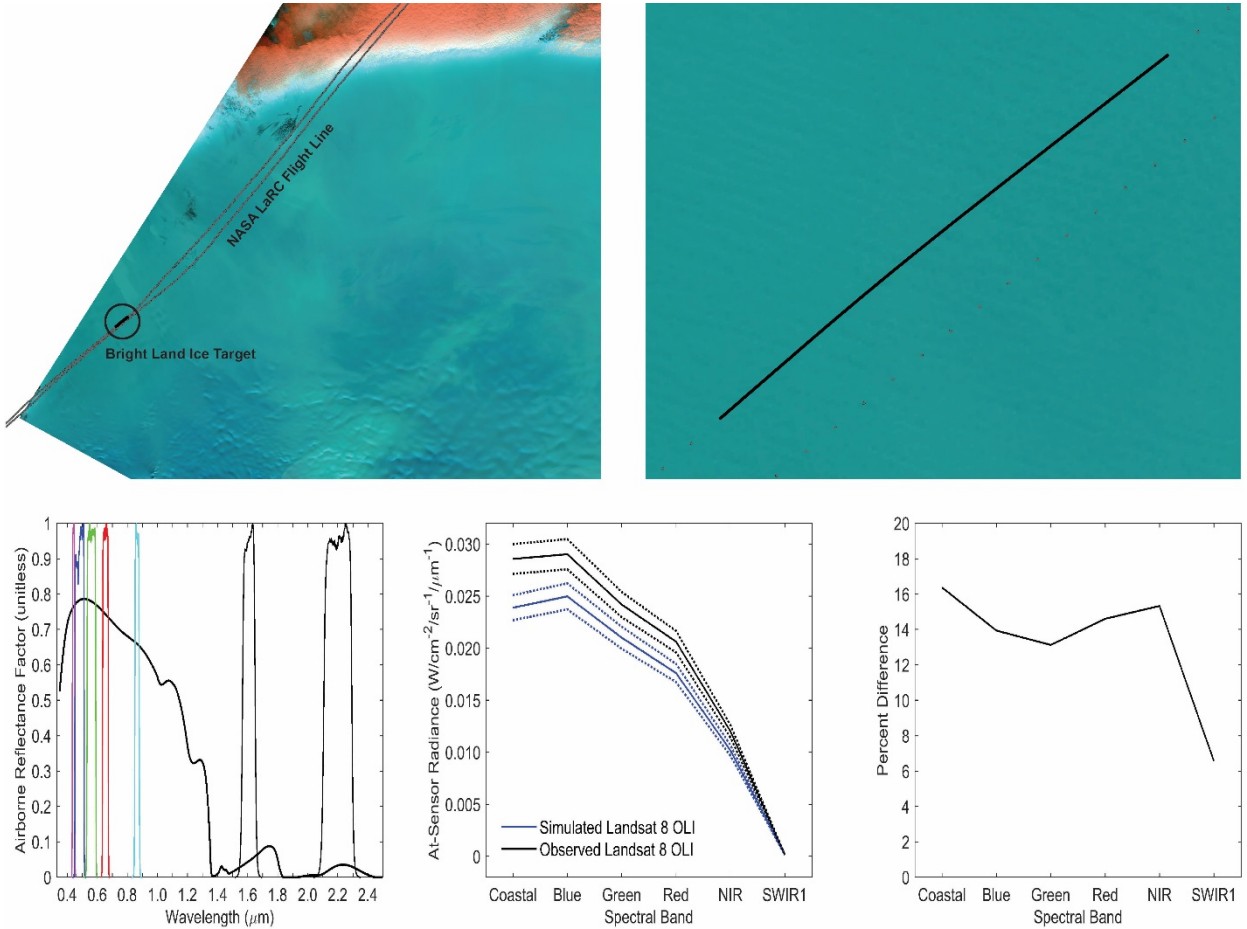

**Figure 15.** MODTRAN simulated at-sensor radiance for coincident Landsat 8 OLI imaging of the bright land ice target using the airborne nadir viewing VSWIR spectrometer reflectance spectrum. The top left panel shows the Landsat 8 OLI image acquisition on 29 July 2015 with the bright land ice target (black line within the black circle), and the NASA LaRC flight line (grey line). The top right panel shows the flight segment (black line) measurements over bright land ice that was used to simulate Landsat 8 OLI at-sensor radiance. The bottom three panels from left to right show (1) Landsat 8 OLI visible, NIR, and SWIR1/2 relative spectral response functions plotted over the bright land ice target airborne reflectance spectrum; (2) a comparison of convolved simulated and observed Landsat 8 OLI at-sensor radiance for the bright land ice target using the average of 24 airborne land ice spectra, and the average of 24 closest Landsat pixels. The dotted lines indicate the within 5% measurement requirement for both Landsat 8 OLI (absolute calibration) and the airborne nadir viewing VSWIR spectrometer (relative calibration); and (3) the percent difference [percent difference= (observed - simulated) / observed] between simulated and observed Landsat 8 OLI at-sensor radiance. Note, at-sensor radiance for Landsat 8 OLI was not simulated for the SWIR2 relative spectral response function based on NASA LaRC BK-7 window transmission uncertainty beyond 2.0 µm.





**Table 1.** Input satellite and AERONET water vapor products for MODTRAN at-sensor simulations of bright land ice for the nadir viewing VSWIR spectrometer.

| Observing System | Retrieval Name | Product | Temporal Resolution | Spatial Resolution | Distance* (km) |
|---|---|---|---|---|---|
| MODIS Aqua | Atmospheric_Water_Vapor_Low | V006, MYD08_D3 | Daily | 1º x 1º | 44.61 |
| MODIS Aqua | Atmospheric_Water_Vapor | V006, MYD08_D3 | Daily | 1º x 1º | 44.61 |
| MODIS Terra | Atmospheric_Water_Vapor | V006, MOD08_D3 | Daily | 1º x 1º | 44.61 |
| AIRS | Atmospheric Water Vapor | AIRS3STD | 12 hour | 2.3 km | 24.13 |
| Thule AB CIMEL | Water | Version 3 | <Hourly | Point-based | 156.35 |

*refers to distance to bright land ice target

