# Peer review of "Radiometric calibration of a non-imaging airborne spectrometer to measure the Greenland Ice Sheet surface"

_Atmospheric Measurement Techniques, 2018_

## Referee Comment (RC1) · Anonymous Referee #2 · 29 Oct 2018

This paper reports on the use of airborne measurements of shortwave spectral reflected radiance and incident spectral irradiance to aid in ICESat-2 validation. Much of the effort reported in this paper is on calibration and characterization of the instruments and on comparison of measurements with radiative transfer simulations over a full dynamic range from darkest to brightest surfaces. In the spirit of reporting the details behind the calibration, testing and analysis, and in the context of this journal, I think this is a publishable manuscript. However, I think the authors must take the time to revise the manuscript based on the numerous comments that follow. I prefer not to segregate between major and minor revisions; the number of comments suggest that in total, the requested revision is major.

[Figure]

Some general comments: the number of acronyms used in the paper make reading very difficult. At times one would benefit from a key to keep track of the multitude of acronyms, many of which are nearly identical. There is also widespread application of jargon. For example, "at-sensor" can be stricken from the entire paper since only at-sensor radiance is investigated. Benchmark and benchmarking is used repeatedly – never once is it defined. Moreover, I would argue that the comparisons between measured and modeled spectra is misinterpreted. In the paper, it is defined as accuracy – it most certainly is not. If detailed error analysis of the simulations is conducted I think the authors will find significant overlap in their respective uncertainties. The conclusion will probably be that they agree to within their uncertainties, likely to be on the order of about 10%, but variable across the spectrum. It is on that basis that conclusions must be drawn. Then they can address that if 2% uncertainties are required, what can be learned? The authors must make revisions along these lines; a model uncertainty budget must be conducted.

Here are the detailed comments, with page and line numbers listed for most:

1. p. 2, l. 30: "benchmark"; Instead of using a term like this it is probably better to be specific about how you intend to use your simulations of spectral radiance and irradiance. As it currently reads, the measurements are to be tested against the modeled "truth". This begs the question, why are the measurements even needed?

2. p. 3, l. 3: what are "profile measurements"?

3. p. 3, l 29: "...within infrared wavelengths..." I think you mean "...in the near-infrared..." Also, you should be more specific and identify water as the absorber.

4. p. 4, l. 7: You should state the fields-of-view. I note that it is done later but it does not hurt to list them here.

5. p. 4, l. 14: "at-sensor" is jargon best used when there is some ambiguity about the vertical location of the radiance: water- or surface-leaving, etc. In this context, where

else can a sensor measure radiance other than at-sensor?!

6. p. 4, l. 18: spectral response functions of what?

7. p. 4 l. 15-18: the list of model input in confusing. Why are standard aerosol profiles and CIMEL measurements (of what? Presumable aerosol optical depths?) both used? What are the assumed state parameters? US standard atmosphere?

8. p. 4 l. 28: "gold standard" is jargon. What characteristics make it the best comparison reference? Does it have the appropriate near-infrared channels for this study?

9. p. 4 l..32: "sunlight" should be "sunlit" and change to "... regions with greater than 5..."

10. p. 4, l. 33: "... challenges that are a result of ..." Aren't the items that follow actually the challenges? If not, then state what the challenges are.

11. p. 4 l. 34: "longer path": longer than what? "greater atmospheric refraction": greater than what?

12. p. 5, l. 23: what is the remote cosine receptor? A transmissive or reflective diffuser? Integrating sphere?

13. p. 6, l. 13: The SWIR1 and SWIR2 detectors had not yet been identified. After going back, I think I know what they are – you need to state explicitly. This makes already confusing notation even more confusing: sometimes the acronyms use in this paper are longer that what they are meant to represent!

14. p. 6, l. 13: I have a hard time understanding what "the entire airborne mission that included a dark current subtraction during each flight" is supposed to mean.

15. p. 6, l. 28-29: What was optimized? Gains?

16. p. 6, l. 28: What is the NIST source? Lamp? Lamp plus integrating sphere? And is it really a NIST source or is it NIST-traceable?

17. p. 7. l. 2: point out the water vapor absorption bands evident in the stability curve of figure 2.

18. pp. 6-7: The figures in figure 2, especially the linearity curve in the upper left needs to be better explained, either in the text or the caption. The "optimization" in the abscissa is not explained.

19. p. 7, l. 4: Is 1 nm resolution the full-width half-maximum of the slit function, that is, spectral resolution? Or is it sampling resolution? And the wavelength precision of 2%: is that 2% of the wavelength scale (for example, 20 nm at 1000 nm, terrible) or 2% of the sampling resolution (2% of 1 nm, very good). Why not remove such ambiguity and list the precision in absolute units, nm?! And finally: the instrument spectral and sampling resolutions must be stated earlier in the text.

20. p. 7, l. 6. Now using Fieldspec 3 or Pro is extremely confusing and requires a scorecard or flipping back to see which instrument is which. Unless the reader works for ASD or used their products, they won't care. Please use the same identifying notation (how about simpoly zenith and nadir spectrometers?!) throughout?

21. p. 7, l. 7: What is the significance of "a PANalytical company"?

22. p. 7. L. 8-9: "less than 2% for 1 nm resolution". Same comment as above.

23. p. 7, l. 22: "per manufacture specifications." Do you mean "in agreement with manufacture specifications"?

24. p. 7, l. 24-26: I don't understand this sentence. Are you saying that window transmission should be appreciably larger than instrument stability? But this leads to a more important question, relevant to the previous paragraph: why wasn't a calibration made with the window/dome in place?

25. p. 7, last paragraph: The listed accuracies are really uncertainties rather than accuracies. How were they derived? Was a correction for window transmittance made? How was the solar zenith angle factored in to the uncertainty? For the zenith measure-

ments (presumably using the cosine receptor) that will likely be the largest source of error, especially if the platform was not actively leveled and in the Arctic where solar zenith angle is quite high.

26. p. 8, l. 6: Is it really an "in-flight radiance calibration strategy" or a strategy to optimize gain and integration time settings? If it is really in-flight-calibration, please explain what standard sources or detectors you are using during flight.

27. p. 8, l. 9: same as above.

28. p. 8, second paragraph: How about the effect of aircraft attitude on solar zenith error and its impact on downwelling irradiance error? (never mind; this is a topic for the next section, but it is not cover there either.)

29. p. 8, last paragraph: what is the threshold for setting the flag?

30. p. 9, l 8: You should probably say more about "cloud contamination". Why do clouds limit the retrieval of surface properties from spectral reflectance measurements? After all, since you are measuring incident irradiance (at flight altitude) it might seem like clouds can be accommodated.

31. p. 9, l. 9: "calibration strategy" again. See previous comment.

32. p. 9, second paragraph: I don't understand this – it seems like it defeats the entire purpose of measuring incident irradiance!

33. p. 9, l. 24: By "direct path" do you mean directly transmitted irradiance? A horizontal translation of the aircraft will be insignificant compared to pitch and roll offsets! I have yet to see this considered, or the angular response of the cosine receptor presented.

34. p. 9, l. 9: The mysterious NIST-traceable source has yet to be identified.

35. Section 3.3.2: I cannot tell if the angular calibrations are for azimuthal or zenith response or both. This is very important. On the other hand, if directly transmitted

solar irradiance cannot be sensed with the orange can in place it does not matter!

36. p. 11, l. 15-16: What is "direct cloud-sky"? Does that imply a broken cloud field? Cloud optical depth low enough such that direct transmittance is appreciable?

37. p. 11, l. 17: Listing times suggests the window can be nothing other than temporal.

38. P. 11, l. 20: Parabolic corrections for what?

39. p. 11, l. 25: Discriminate diffuse sky conditions from what? Or do you mean "identify diffuse sky conditions"?

40. p. 11, l. 31: "...discriminator for sky conditions more broadly..." More broadly than what? I cannot understand, either from the text or figure 7, how the zenith measurements can discriminate between different types of diffuse sky sources. Considerably more explanation is required.

41. Figure 7: I assume the radiation model was plane-parallel; how did you account for the complex cloud geometries? Nothing was said about the modeling in either the text or the captions.

42. p. 12, l. 5, first sentence in paragraph: I think you need to be more specific about what the science is. I assume you are trying to retrieve surface reflectance. You must state this specific application of these measurements. Many other applications do not require surface reflectance validation.

43. p. 12, l. 26 (and many other places): Again, the use of the term "benchmark" comes with no qualification. What do you intend to do with the MODTRAN simulations! Of course, compare them to measurements, but toward what end?

44. p. 12, l. 30: Not sure how you can check for changing solar illumination if you cannot see the sun with the orange can!

45. p. 13, eq. 3: This equation does not contain the atmospheric transmittance from TOA to flight altitude or from flight altitude to ground. Is that what is meant by "apparent"

reflectance? After finishing the paper, I don't see that this was ever considered. It is either flawed or you need to explain how it was applied.

46. p. 13, l. 21: "…to parameterize MODTRAN…" makes no sense. Do you mean "…to initialize MODTRAN…"?

47. p. 14, l. 6: Those are usually called slit functions, not spectral response functions. And finally, a mystery solved (from fig. 11): spectral resolution is 3 nm and 10 nm; 1 nm is sampling resolution. Please state this in the text.

48. p. 14, l. 15: Again, poor or confusing usage: "…successfully constrain MODTRAN …" And as before, do you mean initialize? Or are you really constraining MODTRAN output over a range of input? The caption in Fig. 12 provides no clue. And once again, I think "parameterize" is misused again in the following line.

49. P. 16, l. 4-6: Is accuracy defined to be the relative difference of measured reflectance from MODTRAN reflectance? This is not accuracy! It is just that, a difference between simulation and measurement. Do the simulations have no error? And if not, why are you even trying to make these challenging measurements!

50. p. 16, l. 11: I was hoping this paragraph would quantify uncertainty in the model; alas, it does not. Without it is impossible to assess the significance of comparisons in fig. 14 and in the OLI comparisons presented on the next paragraph.

51. p. 17, l 13: Spectra is the plural of spectrum, not spectrums.

52. P. 17, l. 20-21: Again, what is listed in uncertainty, not accuracy.
* * *

---

## Author Comment (AC1) · 18 Dec 2018

The authors appreciate Referee #2's time and effort to review and provide comments on this AMTD paper. The authors are confident that appropriate responses have been drafted to address the comments offered and expect that such revisions have improved the paper's clarity, conclusions, and impact.

The authors realize that the number of acronyms is substantial. We were aware of this when drafting the paper but felt that the acronyms used are found elsewhere in the remote sensing and cryosphere published literature. That said, we re-evaluated the acronyms and reduced where appropriate to enhance the paper's readability and clar-

ity. We also reduced the jargon and removed the term 'benchmarking' from the paper. We also replaced the term MODTRAN 'simulation' to MODTRAN 'prediction' throughout the text to maintain consistency with other published literature on applications of MODTRAN in airborne science.

The authors employed well defined and published methods for evaluating airborne remote sensing VSWIR instrument measurement performance. Use of a radiative transfer model such as MODTRAN is common practice in airborne science, so the authors are not quite certain what it is that they misrepresented. Along those lines, we do accept that the 'accuracy' term and results needed to be re-evaluated against what is reported, and thus, was changed to 'uncertainty' instead. We address this during responses to specific comments below. We do not understand the reference to the 10% uncertainty in the comments. We are careful in the paper to acknowledge variability in uncertainty across the spectrum at an 80% atmospheric transmission level. Our in-flight calibration is absolute because we measured bright and dark targets and then modeled the atmosphere under approximately the same solar illumination conditions with actual observations from the Thule Air Base CIMEL, and well validated satellite sensor retrievals. We used expert knowledge to define the MODTRAN parameterization for the Arctic atmosphere we were measuring and flying through over the Greenland Ice Sheet. We closely examined the below comments to further clarify our approach and results.

The 2% metric comes from the published literature on Polar ice sheet remote sensing using optical instruments. In the paper, we are drawing attention to the fact that it will likely be difficult to achieve that level of measurement uncertainty. Our approach incorporated robust instrument radiometric calibration methods and traceable SI standards coupled with state of the art atmospheric radiative transfer modeling when making these airborne measurements. Without such approach, it would be hard to determine whether a 2% measurement uncertainty of the ice sheet surface is achievable from airborne and/or spaceborne remote sensing.

Regarding model uncertainty and the error budget analysis, if we were to conduct such an exercise, we would have to conduct a sensitivity analysis of MODTRAN parameters spanning both minimum and maximum ranges of each input. This is a separate effort and possible paper that goes beyond the scope of this communication. Error budget analyses of atmospheric radiative transfer models remain largely incomplete across the global domain, so the authors feel that this request is holding this paper to a standard that has yet to be established in the broader remote sensing community and literature. Introducing uncertainty into MODTRAN predictions would reduce the importance and effort in this paper to retrieve actual atmospheric observations during flight and evaluate the measurement performance of the nadir viewing spectrometer.

Please find our responses to specific comments below along with author changes to the paper.

1. p. 2, l. 30: "benchmark"; Instead of using a term like this it is probably better to be specifizc about how you intend to use your simulations of spectral radiance and irradiance. As it currently reads, the measurements are to be tested against the modeled "truth". This begs the question, why are the measurements even needed?

Author Response: We removed the term 'benchmarking' from the text and instead, now refer to the process of evaluating MODTRAN predictions of radiance against VSWIR spectrometer measurements as a 'comparison'. Yes, this is correct. We are treating MODTRAN for this analysis as 'truth'. MODTRAN radiative transfer predictions are approximations that helped our team (1) evaluate VSWIR spectrometer measurement performance over the Greenland Ice Sheet; and (2) reproduce solar and atmospheric physical processes leading to the remote sensing measurements of reflected surface radiance. We need both remote sensing measurements and radiative transfer models to conduct Greenland Ice Sheet science and applications.

2. p. 3, l. 3: what are "profile measurements"?

Author Response: Non-imaging profile measurements are defined as along-track radiance spectra of the surface directly below the aircraft within the airborne spectrometer's Instantaneous Field-of-View (IFOV). This sentence has been added to the paper.

3. p. 3, l 29: "...within infrared wavelengths..." I think you mean "...in the near infrared..." Also, you should be more specific and identify water as the absorber.

Author Response: The remote sensing literature is often ambiguous on whether the ice absorption feature officially falls within the near infrared or shortwave infrared wavelength region. Thus, the authors used infrared wavelengths more generally as it is technically correct and does not place preference. However, the authors agree with the Referee and inserted 'near' before infrared. The authors also added absorption by liquid water upon recommendation.

4. p. 4, l. 7: You should state the iňÄelds-of-view. I note that it is done later but it does not hurt to list them here.

Author Response: Corrected.

5. p. 4, l. 14: "at-sensor" is jargon best used when there is some ambiguity about the vertical location of the radiance: water- or surface-leaving, etc. In this context, where else can a sensor measure radiance other than at-sensor?!

Author Response: We removed the at-sensor jargon from the paper.

6. p. 4, l. 18: spectral response functions of what?

Author Response: The spectral response functions for the nadir viewing spectrometer. We inserted text to clarify this phrase.

7. p. 4 l. 15-18: the list of model input in confusing. Why are standard aerosol profiles and CIMEL measurements (of what? Presumable aerosol optical depths?) both used? What are the assumed state parameters? US standard atmosphere?

Author Response: We clarified the list of MODTRAN inputs. The rationale and specific parameters are described in detail later in the paper under the Methods section. A

sub-Arctic geographical and seasonal atmosphere was used as the state parameters, and CIMEL measurements of aerosol optical depth and columnar water vapor were used to parameterize MODTRAN predictions.

8. p. 4 l. 28: "gold standard" is jargon. What characteristics make it the best comparison reference? Does it have the appropriate near-infrared channels for this study?

Author Response: We removed the jargon. Landsat 8 Operational Land Imager (OLI) is being increasingly used to map and monitor ice sheet surface characteristics and change because its coverage has dramatically increased since its launch in 2013. It is an appropriate reference instrument because of its absolute radiometric calibration and on-orbit performance, its medium resolution multispectral VSWIR measurements, and its high latitude temporal imaging frequency. These points are included at appropriate places throughout the paper. The near infrared channel on OLI was not designed to measure the ice sheet surface, but it is an important measurement nevertheless. Landsat has and continues to be used widely in Polar remote sensing studies.

9. p. 4 l..32: "sunlight" should be "sunlit" and change to "... regions with greater than 5...

Author Response: We replaced sunlight to sunlit. We revised the language regarding Landsat 8 OLI solar elevation imaging requirement. One point of clarity. Landsat 8 OLI is a land mission that images near shore coastal regions. Thus, saying all regions is not consistent with mission imaging requirements because we do not nominally measure Oceans.

10. p. 4, l. 33: "... challenges that are a result of ..." Aren't the items that follow actually the challenges? If not, then state what the challenges are.

Author Response: We removed the challenges terminology to more accurately reflect this statement on ice sheet remote sensing.

11. p. 4 l. 34: "longer path": longer than what? "greater atmospheric refraction":

greater than what?

Author Response: This phrase has been removed as a path length and refraction physics are part of low solar illumination angles.

12. p. 5, l. 23: what is the remote cosine receptor? A transmissive or reflective diffuser? Integrating sphere?

Author Response: We added the following sentence to the text to clarify what a remote cosine receptor is. A remote cosine receptor is diffuser optic that transmits incoming irradiance from an 1800 hemispherical view.

13. p. 6, l. 13: The SWIR1 and SWIR2 detectors had not yet been identified. After going back, I think I know what they are – you need to state explicitly. This makes already confusing notation even more confusing: sometimes the acronyms use in this paper are longer that what they are meant to represent!

Author Response: We clarified what we mean by SWIR1 and SWIR2 on page 5, Section 2.1.

14. p. 6, l. 13: I have a hard time understanding what "the entire airborne mission that included a dark current subtraction during each flight" is supposed to mean.

Author Response: We clarified that the entire airborne mission represents all nine science flights. This statement documents that we used the same spectrometer integration time and gain configuration for each science flight to avoid measurement saturation over the Greenland Ice Sheet under different atmospheric conditions. We converged on the optimal spectrometer measurement configuration during the absolute in-flight calibration experiment described in the text. The dark current subtraction reflects the removal of internal noise from the spectrometer when measuring radiance or irradiance.

15. p. 6, l. 28-29: What was optimized? Gains?

[Figure]

Author Response: The integration time and gains were optimized to the integrating sphere NIST traceable source output. This has been clarified in the text.

16. p. 6, l. 28: What is the NIST source? Lamp? Lamp plus integrating sphere? And is it really a NIST source or is it NIST-traceable?

Author Response: The source is NIST traceable and includes lamps plus integrating sphere. This has been clarified in the text.

17. p. 7. l. 2: point out the water vapor absorption bands evident in the stability curve of figure 2.

Author Response: The water vapor absorption bands have been identified in Figure 2.

18. pp. 6-7: The figures in figure 2, especially the linearity curve in the upper left needs to be better explained, either in the text or the caption. The "optimization" in the abscissa is not explained.

Author Response: The linearity test in upper left panel of Figure 2 has been clarified in the caption and the figure description has been revised. The linearity test has also been revised for clarity in the text itself.

19. p. 7, l. 4: Is 1 nm resolution the full-width half-maximum of the slit function, that is, spectral resolution? Or is it sampling resolution? And the wavelength precision of 2%: is that 2% of the wavelength scale (for example, 20 nm at 1000 nm, terrible) or 2% of the sampling resolution (2% of 1 nm, very good). Why not remove such ambiguity and list the precision in absolute units, nm?! And finally: the instrument spectral and sampling resolutions must be stated earlier in the text.

Author Response: The spectral resolution of the VNIR detector is 3 nm, and the sampling resolution is 1 nm achieved using an order sorting filter. The spectral resolution of the SWIR detectors is 10 nm with 1 nm sampling resolution. The wavelength scale is 2% of 1 nm, very good. We prefer to report in percentages rather than absolute units as percentages normalize for different radiance levels across the VSWIR range.

We have clarified the text on wavelength precision. We have inserted spectral and sampling resolutions earlier in the text, Section 2.1 to be exact.

20. p. 7, l. 6. Now using Fieldspec 3 or Pro is extremely confusing and requires a scorecard or flipping back to see which instrument is which. Unless the reader works for ASD or used their products, they won't care. Please use the same identifying notation (how about simpoly zenith and nadir spectrometers?!) throughout?

Author Response: We removed FieldSpec 3 and Pro from the text other than initial description early in the text, Section 2.1 to be exact.

21. p. 7, l. 7: What is the significance of "a PANalytical company"?

Author Response: We removed PANalytical company from the text.

22. p. 7. L. 8-9: "less than 2% for 1 nm resolution". Same comment as above.

Author Response: We revised to match the language used to address Comment 19.

23. p. 7, l. 22: "per manufacture specifications." Do you mean "in agreement with manufacture specifications"?

Author Response: This statement is what we intended to write, but we did attempt to clarify be adding 'material' to the sentence.

24. p. 7, l. 24-26: I don't understand this sentence. Are you saying that window transmission should be appreciably larger than instrument stability? But this leads to a more important question, relevant to the previous paragraph: why wasn't a calibration made with the window/dome in place?

Author Response: We added sentences to paragraph one of this section to clarify why we were unable to measure transmittance of the nadir viewing optical window. We also clarified why window transmission uncertainty was appreciably larger than instrument stability in paragraph two of Section 3.1.3.

25. p. 7, last paragraph: The listed accuracies are really uncertainties rather than accuracies. How were they derived? Was a correction for window transmittance made? How was the solar zenith angle factored in to the uncertainty? For the zenith measurements (presumably using the cosine receptor) that will likely be the largest source of error, especially if the platform was not actively leveled and in the Arctic where solar zenith angle is quite high.

Author Response: How these uncertainty metrics were derived has been included in the text. We chose not to correct for window transmittance because the uncertainty was within our targeted measurement requirement. The solar zenith problem and its uncertainty is described later in the text including aircraft horizontal stability and low solar illumination angles in the Arctic. It is important to highlight that these measurements were largely experimental, and we describe their intended use and interpretation later in the text.

26. p. 8, l. 6: Is it really an "in-flight radiance calibration strategy" or a strategy to optimize gain and integration time settings? If it is really in-flight-calibration, please explain what standard sources or detectors you are using during flight.

Author Response: We revised the first paragraph in Section 3.2.1 to more accurately describe our absolute in-flight radiometric calibration. With respect to the in-flight description, we consider this absolute because we constrained the upper limits of upwelling spectral radiance under near clear sky conditions while flying over bright and dark targets. Ideally, there would have been a ground campaign to measure reflectance with a standard source, such as a calibrated Spectralon panel, but this was not possible as described in subsequent sections of the text. Our technical approach in this paper is the best attempt given the unavailability of ground measurements in-flight.

27. p. 8, l. 9: same as above.

Author Response: This statement was revised.

28. p. 8, second paragraph: How about the effect of aircraft attitude on solar zenith error and its impact on downwelling irradiance error? (never mind; this is a topic for the next section, but it is not cover there either.)

Author Response: Noted. This comment was addressed in Section 3.2.2.

29. p. 8, last paragraph: what is the threshold for setting the flag?

Author Response: We removed this statement as this paper does not present science flight data products.

30. p. 9, l 8: You should probably say more about "cloud contamination". Why do cloud-slimittheretrievalofsurfacepropertiesfromspectralreflectancemeasurements? After all, since you are measuring incident irradiance (at flight altitude) it might seem like clouds can be accommodated.

Author Response: We added text regarding cloud contamination and described why identifying these conditions is important for this effort to assess the nadir viewing spectrometer's measurement performance.

31. p. 9, l. 9: "calibration strategy" again. See previous comment.

Author Response: We revised this text.

32. p. 9, second paragraph: I don't understand this – it seems like it defeats the entire purpose of measuring incident irradiance!

Author Response: Our objective for measuring solar irradiance has been clarified in the first paragraph of Section 3.2.2.

33. p. 9, l. 24: By "direct path" do you mean directly transmitted irradiance? A horizontal translation of the aircraft will be insignificant compared to pitch and roll offsets! I have yet to see this considered, or the angular response of the cosine receptor presented.

Author Response: Yes, directly transmitted irradiance is what we intended to say. This has been revised. The angular response of the remote cosine receptor was measured and described in Section 3.3.2.

34. p. 9, l. 9: The mysterious NIST-traceable source has yet to be identified.

Author Response: We used NIST traceable sources from GSFC's Optics Lab and Radiometric Calibration Lab.

35. Section 3.3.2: I cannot tell ifthe angular calibrations are for azimuthal or zenith response or both. This is very important. On the other hand, if directly transmitted solar irradiance cannot be sensed with the orange can in place it does not matter!

Author Response: The remote cosine receptor angular calibrations were zenith from 00 to 1800. The text has been clarified.

36. p. 11, l. 15-16: What is "direct cloud-sky"? Does that imply a broken cloud field? Cloud optical depth low enough such that direct transmittance is appreciable?

Author Response: The following sentence has been added to clarify what we mean by direct cloud-sky. 'Direct cloud-sky indicates when clouds are fully obstructing the direct path'.

37. p. 11, l. 17: Listing times suggests the window can be nothing other than temporal.

Author Response: This is correct. The real value of these spectral irradiance measurements is found in the temporal domain. We illustrated this in Figure 8 using a short flight segment.

38. P. 11, l. 20: Parabolic corrections for what?

Author Response: A parabolic correction is used to splice together VNIR and SWIR detectors during the conversion from raw counts to radiance. This text is unnecessary as written and is already described earlier in the paper. We removed this statement from the sentence.

39. p. 11, l. 25: Discriminate diffuse sky conditions from what? Or do you mean "identify diffuse sky conditions"?

Author Response: Characterize sky conditions is a more appropriate statement. We revised this sentence.

40. p. 11, l. 31: "...discriminator for sky conditions more broadly..." More broadly than what? I cannot understand, either from the text or figure 7, how the zenith measurements can discriminate between different types of diffuse sky sources. Considerably more explanation is required.

Author Response: This comment is well taken. We have revised these sentences to articulate that the OrangeCan spectral irradiance measurements can be used to distinguish between diffuse cloud-sky and diffuse clear-sky based on the amount of irradiance received. This can be visually observed in Figure 7 and detected in the time domain with the Figure 8 illustration.

41. Figure 7: I assume the radiation model was plane-parallel; how did you account for the complex cloud geometries? Nothing was said about the modeling in either the text or the captions.

Author Response: No radiation model was used to derive results shown in Figure 7. In terms of MODTRAN predictions, yes, we assumed plane-parallel and only used 29 July airborne measurements acquired under clear-sky conditions. We used the spectral irradiance data to identify dffuse clear-sky measurements. We assumed direct transmitted clear-sky because the upwelling radiance from the surface exhibited little deviation indicating surface homogeneity with no shadowing. This has been clarified in the text.

42. p. 12, l. 5, first sentence in paragraph: I think you need to be more specific about what the science is. I assume you are trying to retrieve surface reflectance. You must state this specific application of these measurements. Many other applica-

tions do not require surface reflectance validation.

Author Response: We have revised this paragraph for clarity of the radiative transfer comparison methodology. At this point, we are not trying to retrieve surface reflectance, only compare the nadir viewing spectrometer's measurement performance against MODTRAN. While most remote sensing applications do not demand surface reflectance validation because pixel classification based on feature space is the objective, our view is that validation should be undertaken when specific surface properties are being retrieved from reflectance. This paper is an important step towards establishing surface reflectance uncertainty from airborne measurements during this mission.

43. p. 12, l. 26 (and many other places): Again, the use of the term "benchmark" comes with no qualification. What do you intend to do with the MODTRAN simulations! Of course, compare them to measurements, but toward what end?

Author Response: We removed the 'benchmark' terminology from the entire paper. The end game is to use MODTRAN to atmospherically-correct airborne reflectance from the nadir viewing spectrometer.

44. p. 12, l. 30: Not sure how you can check for changing solar illumination if you cannot see the sun with the orange can!

Author Response: We agree. This phrase has been removed. This is not what we meant to say.

45. p. 13, eq. 3: This equation does not contain the atmospheric transmittance from TOA to flight altitude or from flight altitude to ground. Is that what is meant by "apparent" reflectance? After finishing the paper, I don't see that this was ever considered. It is either flawed or you need to explain how it was applied.

Author Response: You are correct. The equation as written does not include transmittance. Apparent reflectance is a widely published term in airborne science and in the remote sensing calibration/validation literature. Because we did not measure ground

reflectance, we treated this low altitude apparent reflectance measurement as a substitute because that is the best possible scenario. We replaced equation 3 in the original paper with equations 3,4, and 5 in the authors changes to the paper. We now completely describe the process we used for retrieving apparent reflectance and have included the proper mathematical notation and supporting citations. MODTRAN is used to simulate these atmospheric quantities during the process of predicting radiance. We have clarified and added text for our radiative transfer method.

46. p. 13, l. 21: "...to parameterize MODTRAN..." makes no sense. Do you mean "...to initialize MODTRAN..."?

Author Response: Yes, we mean parameterize MODTRAN. The use of initialize here would indicate a starting value that will likely change during successive calculations. This is not the case when predicting remote sensing radiance from MODTRAN.

47. p. 14, l. 6: Those are usually called slit functions, not spectral response functions. And finally, a mystery solved (from fig. 11): spectral resolution is 3 nm and 10 nm; 1 nm is sampling resolution. Please state this in the text.

Author Response: Within a MODTRAN environment, slit functions are used to integrate finer spectral resolution radiances to coarse resolution radiances. Spectral response functions were derived specifically for the nadir viewing spectrometer. We describe the spectral resolution of the VNIR and SWIR detectors in the text much earlier now.

48. p. 14, l. 15: Again, poor or confusing usage: "...successfully constrain MODTRAN ..." And as before, do you mean initialize? Or are you really constraining MODTRAN output over a range of input? The caption in Fig. 12 provides no clue. And once again, I think "parameterize" is misused again in the following line.

Author Response: We have revised this sentence. We prefer 'parameterize' over initialize, see response to comment 46 for rationale. In the following sentence, we replace parameterize with 'model'. The caption in Fig. 12 has been clarified.

49. P. 16, l. 4-6: Is accuracy defined to be the relative difference of measured reflectance from MODTRAN reflectance? This is not accuracy! It is just that, a difference between simulation and measurement. Do the simulations have no error? And if not, why are you even trying to make these challenging measurements!

Author Response: We agree this is not accuracy, instead, we define as uncertainty. We compared radiances not reflectance. The difference is used to quantify uncertainty between MODTRAN and measured radiances. Model predictions are approximations. Our objective is not to investigate MODTRAN error but rather, assess nadir viewing spectrometer measurement performance against a well-tested radiative transfer model using expert knowledge and actual atmospheric measurements. These challenging measurements advance Polar ice sheet remote sensing while relying on a solid methodological foundation in airborne science.

50. p. 16, l. 11: I was hoping this paragraph would quantify uncertainty in the model; alas, it does not. Without it is impossible to assess the significance of comparisons in fig. 14 and in the OLI comparisons presented on the next paragraph.

Author Response: We address MODTRAN uncertainty in the summary paragraphs prior to responses to specific comments. We are not attempting to assess the significance of comparisons in Fig 14 and for OLI, but rather, draw attention to the fact that more investigation is required based on this paper's results.

51. p. 17, l 13: Spectra is the plural of spectrum, not spectrums.

Author Response: Corrected.

52. P. 17, l. 20-21: Again, what is listed in uncertainty, not accuracy.

Author Response: We agree and have revised accordingly.

Please also note the supplement to this comment:
https://www.atmos-meas-tech-discuss.net/amt-2018-170/amt-2018-170-AC1-

supplement.pdf

**Supplement:**

[revised manuscript text omitted]

**MODIS Aqua: July 29, 2015**

**Landsat 8 OLI: July 29, 2015**

**Bright Target Flight Segment**

**Landsat 8 OLI: July 30, 2015**

**Dark Target Flight Segment**

**LaRC Flight Photo: July 29, 2015**

**Dark Target Camera Frame**

**Figure 9.** The 29 July flight line showing bright and dark target MODTRAN comparison segments for the nadir viewing spectrometer. The upper left panel shows a MODIS Aqua image (false color SWIR, NIR, Green composite) with the UC-12B flight line (grey line). The upper right panel shows a Landsat 8 OLI image (false color SWIR, NIR, Green composite) with the bright Greenland ice target flight segment (black line within the black dotted circle). The lower left panel shows a Landsat 8 OLI image (false color SWIR, NIR, Green composite) with the dark bare rock/soil target flight segment (black line within the white dotted circle). The lower right panel shows a UC-12B high resolution visible camera image (true color Red, Green, Blue composite) frame of the dark bare rock/soil target flight segment. Note, high resolution visible camera images were acquired over Greenland ice during the campaign science flights.

[Figure]

**Figure 10**. Apparent reflectance spectra for bright and dark absolute in-flight targets measured with the nadir viewing spectrometer.

[Figure]

[Figure]

**Figure 11.** Gaussian spectral response functions for the airborne nadir viewing spectrometer. The left panel shows the VNIR detector spectral response, and the right panel shows the SWIR1/2 detector spectral response. Note, FWHM refers to full width half maximum response to a filter value of 1.0 on the center wavelength.

[Figure]

**Figure 12**. The CIMEL plots show the variability of the spectral aerosol optical depth (top panel), aerosol size distribution (middle panel), and water vapor (bottom panel) throughout the day (29 July). The nadir viewing spectrometer acquisition time for the dark bare rock/soil target was 17.00 UTC (decimal time); the bright Greenland ice target was 16.68 UTC. The meteorological range based on the aerosol optical depth at 550 nm was 67.27 km and 95.48 km, respectively.

[Figure]

**Figure 13**. The airborne nadir viewing spectrometer's measurement sensitives to columnar water vapor for bright Greenland ice and dark bare rock/soil targets. A variety of satellite columnar water vapor data products where evaluated for the bright Greenland ice target due to the remoteness of the flight line segment and its proximity to the Thule Air Base CIMEL.

| Deleted: VSWIR |
| Deleted: at-sensor |
| Deleted: Columnar water vapor absorption lines centered at 0.9 μm and 1.13 μm are designated with black dotted lines. |

[Figure]

**Figure 14**. The airborne nadir viewing spectrometer's measurement performance for bright and dark targets as compared against MODTRAN. The top left panel shows a comparison between predicted and measured radiance for bright Greenland ice. The top right panel shows a predicted verse measured comparison for dark bare rock/soil. The bottom left panel describes

5 the percent difference [i.e., percent difference= (measured – predicted ) / predicted ] between predicted and measured nadir viewing spectrometer radiance for bright Greenland ice (blue line). The percent difference for the dark bare rock/soil target is shown in the bottom right panel. The dotted and top thick black lines on both panels signify the measurement requirement and predicted atmospheric transmittance, respectively. The nadir viewing spectrometer's measurement performance beyond 2.0 μm is subject to noise created by UC-12B BK-7 window transmission, and low to relatively low SWIR radiances for

10 both bright and dark targets.

[Figure]

**Figure 15.** MODTRAN predicted radiance for coincident Landsat 8 OLI imaging of the bright Greenland ice target using the nadir viewing spectrometer apparent reflectance spectrum. The top left panel shows the Landsat 8 OLI image acquisition on 29 July 2015 with the bright Greenland ice target (black line within the black circle), and the UC-12B flight line (grey line). The top right panel shows the flight segment (black line) measurements over bright Greenland ice that was used to predict Landsat 8 OLI radiance. The bottom three panels from left to right show (1) Landsat 8 OLI visible, NIR, and SWIR1/2 relative spectral response functions plotted over the bright Greenland ice target apparent reflectance spectrum; (2) a comparison of convolved predicted and measured Landsat 8 OLI radiance for the bright Greenland ice target using the average of 24 airborne Greenland ice spectra, and the average of 24 closest Landsat pixels. The dotted lines indicate the within 5% measurement requirement for both Landsat 8 OLI (absolute calibration) and the airborne nadir viewing spectrometer (relative calibration); and (3) the percent difference [percent difference= (measured − predicted) / measured ] between predicted and measured Landsat 8 OLI radiance. Note, radiance for Landsat 8 OLI was not predicted for the SWIR2 relative spectral response function based on UC-12B BK-7 window transmission uncertainty beyond 2.0 μm.

**Table 1.** Input satellite and AERONET water vapor products for MODTRAN predictions of bright Greenland ice for the nadir viewing spectrometer.

*refers to distance to bright Greenland ice target

| Observing System | Retrieval Name | Product | Temporal Resolution | Spatial Resolution | Distance* (km) |
|---|---|---|---|---|---|
| MODIS Aqua | Atmospheric_Water_Vapor_Low | V006, MYD08_D3 | Daily | 1° x 1° | 44.61 |
| MODIS Aqua | Atmospheric_Water_Vapor | V006, MYD08_D3 | Daily | 1° x 1° | 44.61 |
| MODIS Terra | Atmospheric_Water_Vapor | V006, MOD08_D3 | Daily | 1° x 1° | 44.61 |
| AIRS | Atmospheric Water Vapor | V006, AIRS3STD | 12 hour | 2.3 km | 24.13 |
| Thule AB CIMEL | Water | Version 3 | <Hourly | Point-based | 156.35 |

---

## Referee Comment (RC2) · Anonymous Referee #3 · 21 Dec 2018

Summary The aim of this work is to evaluate the performance of an airborne visible-to-shortwave infrared (VSWIR) spectrometer by comparing observed radiances with the same collocated with the Landsat 8 OLI sensor and with modeled (MODTRAN) upwelling radiances. The VSWIR detector is part of a suite of sensors used to validate a lidar prototype, which was used in preparation of the ICESat-2 laser altimeter mission. The work presented focuses one case study, a flight over bright and dark surfaces in Greenland during summer of 2015.

Overall, the approach presented is the standard procedure for vicarious calibration of an airborne or satellite sensor: observed radiances are compared against modeled

radiances calculated using all possible ancillary information available regarding the state of the atmosphere and surface at the time of the observation. The description of instrumentation, flight plan and modeling and satellite used constitute all the right tools for such assessment. The descriptions of instrument setup and verification along details provided are adequate

However, there are two major issues that need to be properly addressed and this is the reason I think the paper should be returned for major changes and encourage resubmission (or even reject it to provide more time to work on them). First is the atmospheric aerosol data used for the simulation are not adequate. The Aeronet Level 1.0 data used is most likely most likely contaminated by clouds as the extremely very values in figure 12 demonstrated. Even the low values (∼0.15) are considered clean-to-moderate-low concentration of aerosols. Given the large dynamic range in aerosol loading shown in the plot, it renders the computations questionable at this point. It is recommended that Aeronet level 2.0 (version 3) should be used. Second and more importantly, the paper fails to make the case on what is the novel scientific and/or atmospheric technique contribution of the work presented. As it is now, it just reads as a technical report using standard techniques and procedures to carry out a vicarious calibration.

Some minor comments/clarifications requested: Why no lidar information is used for the constraining the atmospheric radiance simulation? It is my understanding that ICESat-2 is not an atmospheric profilers so I assume the lidar airborne version used in this campaign does not have this capability either. I think it would be desirable to clarify why the lidar onboard is not suitable to aerosol applications.

Through the text all references to figures should specifically to what panel the text refers to. Most of the figures have multiple sub-figures and they are not labeled. Please do so.

Figures Figure 1 does not seem to add information, consider removing it. Figures 2

and 3 : not clear figures. Upper right panels all lines look the same have similar colors. Upper left panel: not clear what it is being compared. Please clarify in caption and main text. Bottom panel: not clear the plot means, what do you mean with stability in this case? Figure 9: all 4 figures did not print well. Particularly the right two panels are just not informative because the lack of contrast even when figure is seen in a computer screen. I think the right two panels can be removed. Figure 11: for consistency with other figures, plot wavelengths in the x-axis. Figure 12: Aeronet figures from Aeronet website are not publication quality material Please plot the data with adequate plotting software.

Figure 15: upper right figure has very poor contrast and it does not provide additional information. Consider removing it. Bottom center images: lines are too thin and difficult to tell the different in them.

[Figure]

---

## Author Comment (AC2) · 15 Jan 2019

The authors thank Anonymous Referee #3 for comments provided on this AMTD paper. We are confident that we have addressed the concerns raised by Referee #3 in our response below. We have revised the original manuscript to reflect Referee #3's recommendations. As a result, we feel that the paper is much improved based on further clarification and more description of the scientific and technical approach as well as better print production on figure illustrations and connection to the text.

We would like to clarify that the goal of the airborne VSWIR spectrometer suite was to support NASA's ICESat-2 project in their efforts to evaluate possible green laser pulse

penetration biases into snow and ice; not to validate a lidar prototype (i.e., SIMPL). VSWIR measurements resulting from this airborne mission will help to characterize snow, ice, and liquid water surface optical properties during airborne science flights while also supporting ICESat-2 calibration/validation objectives. This paper is comprehensive in establishing the scientific basis for VSWIR measurements of snow, ice, and liquid water surfaces and their acquisition, and documents both the airborne spectrometer's traceable radiometric calibration and its airborne measurement performance in the Arctic atmosphere using MODTRAN and Landsat 8 OLI references.

We did use Version 3 AERONET data for our MODTRAN predictions of nadir viewing spectrometer and Landsat 8 OLI radiances. We agree that the AERONET component of this paper needed clarification and more description. We have included those revisions in the author changes to the original AMTD manuscript. The authors did use Level 2.0 CIMEL retrievals as inputs to MODTRAN. We mistakenly inserted Level 1.0 CIMEL plots in Figure 12. After further review, we have decided to remove this figure from the paper entirely as there is not much valued added.

The authors prefer to avoid the 'novel' term to describe this research and its contribution to Polar and atmosphere remote sensing science. Atmospheric Measurement Techniques publishes a wide variety of scientific papers on topics surrounding remote sensing and its applications, instrument calibration/validation in both laboratory and field environments, and measurement-model comparisons that incorporate atmospheric measurements of all kinds. Because of the complexities of measuring the Greenland Ice Sheet surface with VSWIR remote sensing in the Arctic atmosphere, particularly airborne, we did rely on well-known and well-vetted laboratory and vicarious calibration/validation methods to quantify uncertainties and measurement sensitivities to atmospheric conditions during flight across a dark-to-bright dynamic range. We have added text to the manuscript that clearly articulates the significance of this paper's contributions in the context of atmospheric measurement techniques. This paper's contributions are best described by the following points:

(1) Airborne VSWIR measurements with this level of traceability and radiometric calibration are sparse in Arctic regions prior to this mission.

(2) Application of MODTRAN radiative transfer to airborne VSWIR remote sensing of the Greenland Ice Sheet within the Arctic atmosphere breaks new ground for demonstrating its atmospheric modeling capability and performance.

(3) Prior to this study, Landsat 8 OLI's measurement performance over the Greenland Ice Surface had remained largely unknown. While this is only one case study, it establishes a reference baseline for quantifying Landsat 8 OLI's measurement uncertainty when compared to coincident airborne observations and MODTRAN predictions of upwelling radiance that include aerosols, gaseous absorption, and columnar water vapor effects.

(4) The paper draws attention to the importance of instrument radiometric calibration when acquiring airborne VSWIR measurements over snow and ice surfaces in Polar atmospheric conditions. It also helps to establish VSWIR measurement uncertainties using a measurement-model comparison approach with the goal of identifying downstream implications for Polar ice sheet remote sensing of VSWIR surface conditions and properties.

Anonymous Referee #3: Why no lidar information is used for the constraining the atmospheric radiance simulation? It is my understanding that ICESat-2 is not an atmospheric profilers so I assume the lidar airborne version used in this campaign does not have this capability either. I think it would be desirable to clarify why the lidar onboard is not suitable to aerosol applications.

Author Response: This paper does not include analysis of photo counting lidar information from SIMPL. Retrieval of aerosol information from ICESat-2 like or SIMPL measurements is not part of this paper and was not considered as an input to our MODTRAN radiative transfer method for evaluating the nadir viewing VSWIR spectrometer's measurement performance. We acknowledge that aerosol information maybe be helpful to the work at hand; however, doing so would reflect a departure from standard vicarious techniques for optical instruments, and thus, we chose not to pursue this effort at the current time.

Anonymous Referee #3: Through the text all references to figures should specifically to what panel the text refers to. Most of the figures have multiple sub-figures and they are not labeled. Please do so.

Author Response: This has been corrected.

Anonymous Referee #3: Figure 1 does not seem to add information, consider removing it.

Author Response: We removed the Figure 1 from the text.

Anonymous Referee #3: Figures 2 and 3 : not clear figures. Upper right panels all lines look the same have similar colors. Upper left panel: not clear what it is being compared. Please clarify in caption and main text. Bottom panel: not clear the plot means, what do you mean with stability in this case?

Author Response: Referee #2 raised concerns with Figures 2 and 3. We revised these figures for clarity of information and interpretation. These changes have been included in the author changes to the original AMTD manuscript. Stability in this case refers to the nadir viewing spectrometer's repeatable radiometric performance when measuring a stable radiance output from a laboratory NIST traceable source as we show.

Anonymous Referee #3: Figure 9: all 4 figures did not print well. Particularly the right two panels are just not informative because the lack of contrast even when figure is seen in a computer screen. I think the right two panels can be removed.

Author Response: We prefer to keep Figure 9 as constructed. It importantly shows bright and dark target flight segments that we evaluated using our radiative transfer comparisons with MODTRAN and Landsat 8 OLI on 29 July 2015. We attempted to improve the contrast of Figure 9 in the author changes to the original AMTD manuscript.

Anonymous Referee #3: Figure 11: for consistency with other figures, plot wavelengths in the x-axis.

Author Response: Agreed. We made this change.

Anonymous Referee #3: Figure 12: Aeronet figures from Aeronet website are not publication quality material Please plot the data with adequate plotting software.

Author Response: We originally thought the AERONET plots would be important to show as we did much screening for cloud contamination and evaluation of particulate size distribution. We think the AERONET Version 3 data is better described in the text and have removed Figure 12 from the text.

Anonymous Referee #3: Figure 15: upper right figure has very poor contrast and it does not provide additional information. Consider removing it. Bottom center images: lines are too thin and difficult to tell the different in them.

Author Response: We agree. We removed the upper right panel in Figure 15 and revised the bottom panels to reflect a higher quality print production.

Please also note the supplement to this comment:
https://www.atmos-meas-tech-discuss.net/amt-2018-170/amt-2018-170-AC2-supplement.pdf

———————————————————

[Figure]

**Supplement:**

[revised manuscript text omitted]

**Comment [CC2]:** Due to a lapse in US Federal Government funding for the Department of Interior – U.S. Geological Survey during the author comments period, the lead author has been unable to access computer files for revising figure print production per Referee #3's comments. However, the authors will be making the recommended changes as part of the possible final revised manuscript. We did make changes to the caption to reflect Referee #3's subpanel labelling suggestions.

[Figure]

**Figure 2.** Laboratory cross-calibrationcalibration of the nadir and zenith viewing spectrometers using the NIST traceable source. Panel (a) shows the zenith viewing spectrometer linearity test result, and panel (b) shows the cross-calibration using the NIST traceable source output. Panel (c) summarizes the difference in response between nadir and zenith viewing spectrometers relative to the achieved stability requirement (dotted lines).

**Comment [CC3]:** Due to a lapse in US Federal Government funding for the Department of Interior – U.S. Geological Survey during the author comments period, the lead author has been unable to access computer files for revising figure print production per Referee #3's comments. However, the authors will be making the recommended changes as part of the possible final revised manuscript. We did make changes to the caption to reflect Referee #3's subpanel labelling suggestions.

[Figure]

[Figure]

**Figure 3.** A measure of light transmission through the BK7 optical window mounted within the OrangeCan. Panel (a) shows the NIST traceable source output with and without the optical window. Panel (b) summarizes wavelength-dependent radiance loss due to window transmissivity.

[Figure]

**Figure 4.** Laboratory characterization of the nadir viewing 1° foreoptic lens point spread function and IFOV using the NIST traceable source output. Results from green (panel a) and NIR (panel b) wavelengths at which SIMPL operates were used to summarize in-IFOV (thick black line within the dotted line boundaries) and near-IFOV widths (gray regions within the dotted lines).

**Comment [CC5]:** Due to a lapse in US Federal Government funding for the Department of Interior – U.S. Geological Survey during the author comments period, the lead author has been unable to access computer files for revising figure print production per Referee #3's comments. However, the authors will be making the recommended changes as part of the possible final revised manuscript. We did make changes to the caption to reflect Referee #3's subpanel labelling suggestions.

[Figure]

**Figure 5.** Laboratory characterization of the zenith viewing remote cosine receptor optic using a NIST traceable point source. Green (panel a) and NIR (panel b) wavelengths at which SIMPL operates were used to summarize the OrangeCan's impact on the remote cosine receptor optic IFOV and measured irradiance.

**Comment [CC6]:** Due to a lapse in US Federal Government funding for the Department of Interior – U.S. Geological Survey during the author comments period, the lead author has been unable to access computer files for revising figure print production per Referee #3's comments. However, the authors will be making the recommended changes as part of the possible final revised manuscript. We did make changes to the caption to reflect Referee #3's subpanel labelling suggestions.

[Figure]

**Figure 6**. Remote cosine receptor field experiment results from 15 December 2015. Four separate solar illumination scenarios are represented with coincident hemispherical-sky and OrangeCan-sky spectral irradiance measurements. Average spectral irradiance for each scenario was calculated using one-second measurement sampling for local time and solar zenith angle (SZA) shown. Solar illumination conditions along the directly transmitted path and zenith diffuse-sky are shown on the right with photographs. Note: The amount of irradiance is dependent on the temporal proximity to solar noon, which on 15 December 2015 was 11:51 EST.

[Figure]

[Figure]

**Figure 7**. Example zenith remote cosine receptor irradiance measurements for a 29 July flight segment. Panel (a) shows zenith irradiance measurements from 0.4 to 1.0 μm. The black lines indicate variability in instantaneous in-flight irradiance for a 5-minute flight segment. The thick red line signifies the baseline minimum irradiance received, a condition that represents diffuse clear-sky to near clear-sky as verified with Figure 6 results. Panel (b) shows zenith integrated irradiance (i.e., sum function) from 0.4 to 1.0 μm for the same 5-minute flight segment. The thick black line indicates temporal variance in zenith integrated irradiance, a measure of sky conditions above the UC-12B aircraft. The dotted line signifies the computed mode (most frequently occurring condition) of zenith integrated irradiance, an indicator of sky condition stability. The red line serves as the minimum zenith integrated irradiance baseline. Using the temporal variance in zenith integrated irradiance, the mode value, and the minimum value, variable sky conditions during flight can be classified and the nadir viewing spectrometer measurements can be filtered for cloud contamination.

**Comment [CC7]:** Due to a lapse in US Federal Government funding for the Department of Interior – U.S. Geological Survey during the author comments period, the lead author has been unable to access computer files for revising figure print production per Referee #3's comments. However, the authors will be making the recommended changes as part of the possible final revised manuscript. We did make changes to the caption to reflect Referee #3's subpanel labelling suggestions.

MODIS Aqua: July 29, 2015

Landsat 8 OLI: July 29, 2015

[Figure]

[Figure]

**Bright Target Flight Segment**

Landsat 8 OLI: July 30, 2015

LaRC Flight Photo: July 29, 2015

**Dark Target Flight Segment**

**Dark Target Camera Frame**

**Figure 8.** The 29 July flight line showing bright and dark target MODTRAN comparison segments for the nadir viewing spectrometer. Panel (a) shows a MODIS Aqua image (false color SWIR, NIR, Green composite) with the UC-12B flight line (grey line). Panel (b) shows a Landsat 8 OLI image (false color SWIR, NIR, Green composite) with the bright Greenland ice target flight segment (black line within the black dotted circle). Panel (c) shows a Landsat 8 OLI image (false color SWIR, NIR, Green composite) with the dark bare rock/soil target flight segment (black line within the white dotted circle). Panel (d) shows a UC-12B high resolution visible camera image (true color Red, Green, Blue composite) frame of the dark bare rock/soil target flight segment. Note, high resolution visible camera images were acquired over Greenland ice during the campaign science flights.

**Comment [CC8]:** Due to a lapse in US Federal Government funding for the Department of Interior – U.S. Geological Survey during the author comments period, the lead author has been unable to access computer files for revising figure print production per Referee #3's comments. However, the authors will be making the recommended changes as part of the possible final revised manuscript. We did make changes to the caption to reflect Referee #3's subpanel labelling suggestions.

[Figure]

**Figure 9.** Apparent reflectance spectra for bright and dark absolute in-flight targets measured with the nadir viewing spectrometer.

[Figure]

[Figure]

**Figure 10.** Gaussian spectral response functions for the airborne nadir viewing spectrometer. Panel (a) shows the VNIR detector spectral response, and panel (b) shows the SWIR1/2 detector spectral response. Note, FWHM refers to full width half maximum response to a filter value of 1.0 on the center wavelength.

**Comment [CC9]:** Due to a lapse in US Federal Government funding for the Department of Interior – U.S. Geological Survey during the author comments period, the lead author has been unable to access computer files for revising figure print production per Referee #3's comments. However, the authors will be making the recommended changes as part of the possible final revised manuscript. We did make changes to the caption to reflect Referee #3's subpanel labelling suggestions.

[Figure]

**Figure 11.** The airborne nadir viewing spectrometer's measurement sensitives to columnar water vapor for bright Greenland ice (panel a) and dark bare rock/soil (panel b) targets. A variety of satellite columnar water vapor data products where evaluated for the bright Greenland ice target due to the remoteness of the flight line segment and its proximity to the Thule Air Base CIMEL.

**Comment [CC10]:** Due to a lapse in US Federal Government funding for the Department of Interior – U.S. Geological Survey during the author comments period, the lead author has been unable to access computer files for revising figure print production per Referee #3's comments. However, the authors will be making the recommended changes as part of the possible final revised manuscript. We did make changes to the caption to reflect Referee #3's subpanel labelling suggestions.

[Figure]

Figure 12. The airborne nadir viewing spectrometer's measurement performance for bright and dark targets as compared against MODTRAN. Panel (a) shows a comparison between predicted and measured radiance for bright Greenland ice. Panel (b) shows a predicted verse measured comparison for dark bare rock/soil. Panel (c) describes the percent difference [i.e. percent difference= (measured – predicted ) / predicted ] between predicted and measured nadir viewing spectrometer radiance for bright Greenland ice (blue line). The percent difference for the dark bare rock/soil target is shown in Panel (c). The dotted and top thick black lines on panels (c) and (d) signify the measurement requirement and predicted atmospheric transmittance, respectively. The nadir viewing spectrometer's measurement performance beyond 2.0 μm is subject to noise created by UC-12B BK-7 window transmission, and low to relatively low SWIR radiances for both bright and dark targets.

Comment [CC11]: Due to a lapse in US Federal Government funding for the Department of Interior – U.S. Geological Survey during the author comments period, the lead author has been unable to access computer files for revising figure print production ... [33]

[Figure]

[Figure]

[Figure]

[Figure]

[Figure]

**Figure 13.** MODTRAN predicted radiance for coincident Landsat 8 OLI imaging of the bright Greenland ice target using the nadir viewing spectrometer apparent reflectance spectrum. Panel (a) shows the Landsat 8 OLI image acquisition on 29 July 2015 with the bright Greenland ice target (black line within the black circle), and the UC-12B flight line (grey line). Panel (b) shows Landsat 8 OLI's visible, NIR, and SWIR1/2 relative spectral response functions plotted over the bright Greenland ice target apparent reflectance spectrum. Panel (c) is the comparison of convolved predicted and measured Landsat 8 OLI radiance for the bright Greenland ice target using the average of 24 airborne Greenland ice spectra, and the average of 24 closest Landsat pixels. The dotted lines indicate the within 5% measurement requirement for both Landsat 8 OLI (absolute calibration) and the airborne nadir viewing spectrometer (relative calibration). Panel (d) is the percent difference [percent difference= (measured – predicted ) / measured ] between predicted and measured Landsat 8 OLI radiance. Note, radiance for Landsat 8 OLI was not predicted for the SWIR2 relative spectral response function based on UC-12B BK-7 window transmission uncertainty beyond 2.0 μm.

**Comment [CC12]:** Due to a lapse in US Federal Government funding for the Department of Interior – U.S. Geological Survey during the author comments period, the lead author has been unable to access computer files for revising figure print production per Referee #3's comments. However, the authors will be making the recommended changes as part of the possible final revised manuscript. We did make changes to the caption to reflect Referee #3's subpanel labelling suggestions.

**Table 1.** Input satellite and AERONET water vapor products for MODTRAN predictions of bright Greenland ice for the nadir viewing spectrometer.

| Observing System | Retrieval Name | Product | Temporal Resolution | Spatial Resolution | Distance* (km) |
|---|---|---|---|---|---|
| MODIS Aqua | Atmospheric_Water_Vapor_ Low | V006, MYD08_D3 | Daily | 1$^o$ x 1$^o$ | 44.61 |
| MODIS Aqua | Atmospheric_Water_Vapor | V006, MYD08_D3 | Daily | 1$^o$ x 1$^o$ | 44.61 |
| MODIS Terra | Atmospheric_Water_Vapor | V006, MOD08_D3 | Daily | 1$^o$ x 1$^o$ | 44.61 |
| AIRS | Atmospheric Water Vapor | V006, AIRS3STD | 12 hour | 2.3 km | 24.13 |
| Thule AB CIMEL | Water | Version 3 | <Hourly | Point-based | 156.35 |

*refers to distance to bright Greenland ice target

| Observing System | Retrieval Name | Product | Temporal Resolution | Spatial Resolution | Distance* (km) |
|---|---|---|---|---|---|
| MODIS Aqua | Atmospheric_Water_Vapor_ Low | V006, MYD08_D3 | Daily | 1$^o$ x 1$^o$ | 44.61 |

| Page 7: [1] Deleted | Crawford, Christopher (Contractor) J | 12/4/18 3:21:00 PM |

spectroscopic

| Page 7: [1] Deleted | Crawford, Christopher (Contractor) J | 12/4/18 3:21:00 PM |

spectroscopic

| Page 7: [1] Deleted | Crawford, Christopher (Contractor) J | 12/4/18 3:21:00 PM |

spectroscopic

| Page 7: [1] Deleted | Crawford, Christopher (Contractor) J | 12/4/18 3:21:00 PM |

spectroscopic

| Page 7: [1] Deleted | Crawford, Christopher (Contractor) J | 12/4/18 3:21:00 PM |

spectroscopic

| Page 7: [1] Deleted | Crawford, Christopher (Contractor) J | 12/4/18 3:21:00 PM |

spectroscopic

| Page 7: [1] Deleted | Crawford, Christopher (Contractor) J | 12/4/18 3:21:00 PM |

spectroscopic

| Page 7: [2] Deleted | Crawford, Christopher (Contractor) J | 11/25/18 11:22:00 PM |

FieldSpec Pro

| Page 7: [2] Deleted | Crawford, Christopher (Contractor) J | 11/25/18 11:22:00 PM |

FieldSpec Pro

| Page 7: [2] Deleted | Crawford, Christopher (Contractor) J | 11/25/18 11:22:00 PM |

FieldSpec Pro

| Page 7: [2] Deleted | Crawford, Christopher (Contractor) J | 11/25/18 11:22:00 PM |

FieldSpec Pro

| Page 7: [2] Deleted | Crawford, Christopher (Contractor) J | 11/25/18 11:22:00 PM |

FieldSpec Pro

| Page 7: [2] Deleted | Crawford, Christopher (Contractor) J | 11/25/18 11:22:00 PM |

FieldSpec Pro

| Page 7: [2] Deleted | Crawford, Christopher (Contractor) J | 11/25/18 11:22:00 PM |

FieldSpec Pro

| **Page 7: [2] Deleted** | **Crawford, Christopher (Contractor) J** | **11/25/18 11:22:00 PM** |

FieldSpec Pro

| **Page 7: [2] Deleted** | **Crawford, Christopher (Contractor) J** | **11/25/18 11:22:00 PM** |

FieldSpec Pro

| **Page 7: [2] Deleted** | **Crawford, Christopher (Contractor) J** | **11/25/18 11:22:00 PM** |

FieldSpec Pro

| **Page 7: [2] Deleted** | **Crawford, Christopher (Contractor) J** | **11/25/18 11:22:00 PM** |

FieldSpec Pro

| **Page 7: [3] Deleted** | **Crawford, Christopher (Contractor) J** | **11/25/18 11:40:00 PM** |

the instrument's response (Figure 2)

| **Page 7: [3] Deleted** | **Crawford, Christopher (Contractor) J** | **11/25/18 11:40:00 PM** |

the instrument's response (Figure 2)

| **Page 7: [3] Deleted** | **Crawford, Christopher (Contractor) J** | **11/25/18 11:40:00 PM** |

the instrument's response (Figure 2)

| **Page 7: [3] Deleted** | **Crawford, Christopher (Contractor) J** | **11/25/18 11:40:00 PM** |

the instrument's response (Figure 2)

| **Page 7: [3] Deleted** | **Crawford, Christopher (Contractor) J** | **11/25/18 11:40:00 PM** |

the instrument's response (Figure 2)

| **Page 7: [3] Deleted** | **Crawford, Christopher (Contractor) J** | **11/25/18 11:40:00 PM** |

the instrument's response (Figure 2)

| **Page 7: [4] Deleted** | **Crawford, Christopher (Contractor) J** | **11/25/18 11:46:00 PM** |

 FieldSpec Pro

| **Page 7: [4] Deleted** | **Crawford, Christopher (Contractor) J** | **11/25/18 11:46:00 PM** |

 FieldSpec Pro

| **Page 7: [4] Deleted** | **Crawford, Christopher (Contractor) J** | **11/25/18 11:46:00 PM** |

 FieldSpec Pro

| **Page 7: [5] Deleted** | **Crawford, Christopher (Contractor) J** | **11/26/18 2:44:00 PM** |

FieldSpec 3

| **Page 7: [5] Deleted** | **Crawford, Christopher (Contractor) J** | **11/26/18 2:44:00 PM** |

FieldSpec 3

| Page 7: [6] Deleted | Crawford, Christopher (Contractor) J | 11/26/18 2:10:00 PM |
|---|---|---|

(a PANalytical company)

| Page 9: [7] Deleted | Crawford, Christopher (Contractor) J | 12/4/18 3:47:00 PM |
|---|---|---|

Thus, the aircraft along-track LOS within the flight path is important to reconcile relative to direct path solar illumination geometry.

| Page 9: [8] Deleted | Crawford, Christopher (Contractor) J | 11/28/18 10:17:00 PM |
|---|---|---|

Science flights that have the potential for an along-track LOS scattering bias will be flagged in the measurement metadata.

| Page 17: [9] Deleted | Crawford, Christopher (Contractor) J | 12/5/18 1:22:00 AM |
|---|---|---|

NASA

| **Page 17: [9] Deleted** | **Crawford, Christopher (Contractor) J** | **12/5/18 1:22:00 AM** |

NASA

| **Page 17: [10] Deleted** | **Crawford, Christopher (Contractor) J** | **12/5/18 1:26:00 AM** |

benchmark

| **Page 17: [11] Deleted** | **Crawford, Christopher (Contractor) J** | **12/5/18 1:33:00 AM** |

 VSWIR

**Page 17: [11] Deleted**      **Crawford, Christopher (Contractor) J**      **12/5/18 1:33:00 AM**

VSWIR

**Page 17: [11] Deleted**      **Crawford, Christopher (Contractor) J**      **12/5/18 1:33:00 AM**

VSWIR

**Page 17: [11] Deleted**      **Crawford, Christopher (Contractor) J**      **12/5/18 1:33:00 AM**

VSWIR

**Page 17: [11] Deleted**      **Crawford, Christopher (Contractor) J**      **12/5/18 1:33:00 AM**

VSWIR

**Page 17: [12] Deleted**      **Crawford, Christopher (Contractor) J**      **11/20/18 2:19:00 PM**

at-sensor

**Page 17: [12] Deleted**      **Crawford, Christopher (Contractor) J**      **11/20/18 2:19:00 PM**

at-sensor

**Page 17: [12] Deleted**      **Crawford, Christopher (Contractor) J**      **11/20/18 2:19:00 PM**

at-sensor

**Page 17: [12] Deleted**      **Crawford, Christopher (Contractor) J**      **11/20/18 2:19:00 PM**

at-sensor

**Page 17: [12] Deleted**      **Crawford, Christopher (Contractor) J**      **11/20/18 2:19:00 PM**

at-sensor

**Page 17: [12] Deleted**      **Crawford, Christopher (Contractor) J**      **11/20/18 2:19:00 PM**

at-sensor

**Page 17: [12] Deleted**      **Crawford, Christopher (Contractor) J**      **11/20/18 2:19:00 PM**

at-sensor

**Page 17: [12] Deleted**      **Crawford, Christopher (Contractor) J**      **11/20/18 2:19:00 PM**

at-sensor

**Page 17: [13] Deleted**      **Crawford, Christopher (Contractor) J**      **12/5/18 1:37:00 AM**

cal/val

**Page 17: [13] Deleted**      **Crawford, Christopher (Contractor) J**      **12/5/18 1:37:00 AM**

cal/val

| Page 17: [13] Deleted | Crawford, Christopher (Contractor) J | 12/5/18 1:37:00 AM |
|---|---|---|

cal/val

| Page 17: [13] Deleted | Crawford, Christopher (Contractor) J | 12/5/18 1:37:00 AM |
|---|---|---|

cal/val

| Page 18: [14] Deleted | Crawford, Christopher (Contractor) J | 12/5/18 1:39:00 AM |
|---|---|---|

remote sensing

| Page 18: [14] Deleted | Crawford, Christopher (Contractor) J | 12/5/18 1:39:00 AM |
|---|---|---|

remote sensing

| Page 18: [15] Deleted | Crawford, Christopher (Contractor) J | 12/5/18 1:40:00 AM |
|---|---|---|

 simulations

| Page 18: [15] Deleted | Crawford, Christopher (Contractor) J | 12/5/18 1:40:00 AM |
|---|---|---|

 simulations

| Page 18: [15] Deleted | Crawford, Christopher (Contractor) J | 12/5/18 1:40:00 AM |
|---|---|---|

 simulations

| Page 18: [15] Deleted | Crawford, Christopher (Contractor) J | 12/5/18 1:40:00 AM |
|---|---|---|

 simulations

| Page 18: [15] Deleted | Crawford, Christopher (Contractor) J | 12/5/18 1:40:00 AM |
|---|---|---|

 simulations

| Page 18: [15] Deleted | Crawford, Christopher (Contractor) J | 12/5/18 1:40:00 AM |
|---|---|---|

 simulations

| Page 18: [15] Deleted | Crawford, Christopher (Contractor) J | 12/5/18 1:40:00 AM |
|---|---|---|

 simulations

| Page 18: [16] Deleted | Crawford, Christopher (Contractor) J | 12/5/18 1:42:00 AM |
|---|---|---|

 VSWIR

| Page 18: [16] Deleted | Crawford, Christopher (Contractor) J | 12/5/18 1:42:00 AM |
|---|---|---|

 VSWIR

| Page 18: [16] Deleted | Crawford, Christopher (Contractor) J | 12/5/18 1:42:00 AM |
|---|---|---|

 VSWIR

| Page 18: [17] Deleted | Crawford, Christopher (Contractor) J | 12/5/18 1:43:00 AM |
|---|---|---|

 NASA

| Page 18: [17] Deleted | Crawford, Christopher (Contractor) J | 12/5/18 1:43:00 AM |
|---|---|---|

NASA

| Page 18: [18] Deleted | Crawford, Christopher (Contractor) J | 12/5/18 1:44:00 AM |
|---|---|---|

The accuracy of

| Page 18: [19] Deleted | Crawford, Christopher (Contractor) J | 12/5/18 1:46:00 AM |
|---|---|---|

VSWIR spectra

| Page 18: [20] Deleted | Crawford, Christopher (Contractor) J | 12/5/18 1:48:00 AM |
|---|---|---|

NASA

| Page 18: [21] Deleted | Crawford, Christopher (Contractor) J | 12/5/18 1:48:00 AM |
|---|---|---|

simulated

| Page 18: [22] Deleted | Crawford, Christopher (Contractor) J | 12/5/18 1:49:00 AM |
|---|---|---|

parameters

| Page 20: [23] Formatted | Christopher Crawford | 1/13/19 11:35:00 AM |
|---|---|---|

Highlight

| Page 20: [23] Formatted | Christopher Crawford | 1/13/19 11:35:00 AM |
|---|---|---|

Highlight

| Page 20: [23] Formatted | Christopher Crawford | 1/13/19 11:35:00 AM |
|---|---|---|

Highlight

| Page 20: [23] Formatted | Christopher Crawford | 1/13/19 11:35:00 AM |
|---|---|---|

Highlight

| Page 20: [24] Deleted | Christopher Crawford | 1/13/19 11:43:00 AM |
|---|---|---|

| Page 20: [24] Deleted | Christopher Crawford | 1/13/19 11:43:00 AM |
|---|---|---|

| Page 20: [25] Deleted | Crawford, Christopher (Contractor) J | 12/5/18 9:51:00 AM |
|---|---|---|

VSWIR

| Page 20: [25] Deleted | Crawford, Christopher (Contractor) J | 12/5/18 9:51:00 AM |
|---|---|---|

VSWIR

| Page 20: [25] Deleted | Crawford, Christopher (Contractor) J | 12/5/18 9:51:00 AM |
|---|---|---|

VSWIR

| Page 20: [25] Deleted | Crawford, Christopher (Contractor) J | 12/5/18 9:51:00 AM |
|---|---|---|

VSWIR

| Page 20: [25] Deleted | Crawford, Christopher (Contractor) J | 12/5/18 9:51:00 AM |
|---|---|---|

VSWIR

| Page 20: [26] Formatted | Christopher Crawford | 1/14/19 10:13:00 PM |
|---|---|---|

Highlight

| Page 20: [26] Formatted | Christopher Crawford | 1/14/19 10:13:00 PM |
|---|---|---|

Highlight

| Page 20: [26] Formatted | Christopher Crawford | 1/14/19 10:13:00 PM |
|---|---|---|

Highlight

| Page 20: [26] Formatted | Christopher Crawford | 1/14/19 10:13:00 PM |
|---|---|---|

Highlight

| Page 20: [26] Formatted | Christopher Crawford | 1/14/19 10:13:00 PM |
|---|---|---|

Highlight

| Page 20: [26] Formatted | Christopher Crawford | 1/14/19 10:13:00 PM |
|---|---|---|

Highlight

| Page 20: [27] Deleted | Crawford, Christopher (Contractor) J | 12/5/18 9:53:00 AM |
|---|---|---|

NASA

| Page 20: [28] Deleted | Christopher Crawford | 1/13/19 12:11:00 PM |
|---|---|---|

| Page 20: [29] Deleted | Christopher Crawford | 1/13/19 12:11:00 PM |
|---|---|---|

| Page 20: [30] Deleted | Crawford, Christopher (Contractor) J | 12/5/18 10:10:00 AM |
|---|---|---|

GSFC

GSFC

| Page 20: [30] Deleted | Crawford, Christopher (Contractor) J | 12/5/18 10:10:00 AM |

GSFC

| Page 24: [31] Deleted | Christopher Crawford | 1/12/19 12:48:00 AM |

[Figure]

**Figure 1**. The zenith mounted OrangeCan on top of the NASA LaRC UC-12B aircraft. The remote cosine receptor RCR optic was mounted inside directly under a BK7 optical window to characterize sky conditions during flight.

| Page 34: [32] Deleted | Christopher Crawford | 1/12/19 12:51:00 AM |

**Figure 12**. The CIMEL plots show the variability of the spectral aerosol optical depth AOD (top panel), aerosol size distribution (middle panel), and water vapor (bottom panel) throughout the day (29 July). The nadir viewing spectrometer VSWIR acquisition time for the dark bare rock/soil target wasis 17.00 UTC (decimal time); the bright land Greenland ice target was 16.68 UTC. The meteorological range based on the aerosol optical depth at 550 nm $AOD_{550}$ was is 67.27 km and 95.48 km, respectively.

| Page 36: [33] Commented | Christopher Crawford | 1/12/19 1:32:00 PM |
|---|---|---|

Due to a lapse in US Federal Government funding for the Department of Interior – U.S. Geological Survey during the author comments period, the lead author has been unable to access computer files for revising figure print production per Referee #3's comments. However, the authors will be making the recommended changes as part of the possible final revised manuscript. We did make changes to the caption to reflect Referee #3's subpanel labelling suggestions.

| Page 37: [34] Deleted | Christopher Crawford | 1/13/19 12:24:00 PM |
|---|---|---|

The top right panel shows the flight segment (black line) measurements over bright Greenland ice that was used to simulate predict Landsat 8 OLI at-sensor radiance. The bottom three panels from left to right show (1